# Hysteresis of idealized, instability-prone outlet glaciers under variation of pinning-point buttressing

Johannes Feldmann[1], Anders Levermann[1,2,3], and Ricarda Winkelmann[1,2]

[1]Potsdam Institute for Climate Impact Research (PIK), Potsdam, Germany
[2]Institute of Physics, University of Potsdam, Potsdam, Germany
[3]LDEO, Columbia University, New York, USA

**Correspondence:** Johannes Feldmann (johannes.feldmann@pik-potsdam.de)

**Abstract.** Ice rises or ice rumples act as ice-shelf pinning points that can have an important role in regulating the ice discharge of marine outlet glaciers. As an example, the observed recent gradual ungrounding of the ice shelf of West Antarctica's Thwaites Glacier from its last pinning points likely diminished the buttressing effect of the ice shelf and thus contributed to the destabilization of the outlet. Here we use an idealized experimental setting to simulate the response of a marine outlet glacier resting on a landward down-sloping (retrograde) bed to a step-wise ungrounding of its ice shelf from a topographic high and a subsequent re-grounding. We show that the glacier retreat down the retrograde bed, induced by the loss in pinning-point buttressing, can be unstable and irreversible given a relatively deep subglacial bed depression. In this case, glacier retreat and re-advance show a hysteretic behavior and if the bed depression is sufficiently deep, the glacier does not recover but remains locked in its retreated state. Conversely, reversibility requires a sufficiently shallow bed depression. Based on a simple flux balance analysis, we argue that the combination of a deep bed depression and limited ice-shelf buttressing hampers grounding-line re-advance due to the dominant and highly non-linear influence of the bed depth on the ice discharge across the grounding line. We conclude that outlets that rest on a deep bed depression and are weakly buttressed, such as Thwaites Glacier, are more susceptible to abrupt and irreversible retreat than stronger buttressed glaciers on more moderate retrograde slope, such as Pine Island Glacier. Our findings further suggest that the (ir)reversibility of large-scale grounding-line retreat may be strongly affected by calving-front migration and associated changes in ice-shelf buttressing.

## 1 Introduction

Where the floating ice shelves ground on locally elevated seabed, ice rises or ice rumples form. Such ice-shelf pinning points are scattered along major parts of the Antarctic coast (Matsuoka et al., 2015, Fig. 3). Many of them provide enough basal drag to exert a buttressing force on the upstream grounded tributaries that feed the ice shelves (pinning-point induced buttressing), and thus act as a regulator of outlet-glacier ice discharge (Goldberg et al., 2009; Gagliardini et al., 2010; Favier et al., 2012; Berger et al., 2016; Reese et al., 2018). For instance, the ungrounding of East Antarctica's Brunt Ice Shelf from a pinning point in the course of a calving event in the early 1970s has been suggested to have led to a quasi-instantaneous twofold increase in the ice shelf's flow speed (Gudmundsson et al., 2017). Crary Ice Rise of the Ross Ice Shelf has been estimated to exert a resistance that accounts for about half of the buttressing forces acting on the upstream Whillans Ice Stream (MacAyeal et al., 1987).

In-depth numerical modeling studies have investigated, e.g., the speed-up of Larsen C Ice Shelf in response to a reduction in pinning-point buttressing (Borstad et al., 2013), the effect of pinning points on marine outlet glacier stability and grounding line (GL) dynamics in the region between Lazarev and Roi Baudouin ice shelves (Favier et al., 2016), the influence of surface accumulation rate and ocean perturbations on ice-rise divide migration at Ekström Ice Shelf (Schannwell et al., 2019) or the hysteretic evolution of ice rises and rumples under variations of sea level and basal friction in idealized simulations (Henry et al., 2022). Evidence from paleo records suggests that ice-rise formation generally played an important role in Antarctica's glacial history, interfering with the cyclic retreat and advance of the ice sheet (Matsuoka et al., 2015).

While pinning points have the potential to restrain upstream ice flow substantially, ice-shelf buttressing can also emerge from horizontal shearing at the lateral margins of an ice shelf, if it is confined by slowly moving grounded ice or the bed topography (confinement-induced buttressing; Dupont and Alley, 2005; Gudmundsson et al., 2019). In Antarctica, many ice shelves provide a combination of pinning-point induced buttressing and confinement-induced buttressing to the upstream ice flow. Examples range from relatively small, narrow ice shelves with few and small pinning points, like West Antarctica's Pine Island Ice Shelf (Arndt et al., 2018), to the two largest and widest of the ice sheet that incorporate numerous and partly very large pinning points, i.e., Ross and Filchner-Ronne ice shelves (Matsuoka et al., 2015; Still et al., 2019).

Observations over the last two decades indicate that strong basal melting has thinned the ice shelves of West Antarctica's Amundsen Sea sector including the ice shelves of Pine Island and Thwaites glaciers (Paolo et al., 2015; Shepherd et al., 2018; De Rydt et al., 2021; Joughin et al., 2021). The associated buttressing reduction resulted in enhanced ice discharge, speed-up and GL retreat of the two outlets (Konrad et al., 2018; Milillo et al., 2019; De Rydt et al., 2021; Rignot et al., 2022). Part of this buttressing loss was due to the ice shelves' ungrounding from several of their pinning points in the course of their thinning (Tinto and Bell, 2011; Arndt et al., 2018; Miles and Bingham, 2024). The ice shelf of Thwaites Glacier continues to gradually lose contact to its last remaining pinning points (MacGregor et al., 2012; Rignot et al., 2014; Alley et al., 2021; Benn et al., 2022) and could unpin completely in less than a decade (Wild et al., 2022). This might further decrease pinning-point buttressing and contribute to the imbalance of the glacier also in the future (Bett et al., 2023), although to a possibly limited extent (Gudmundsson et al., 2023). In fact, the numerical simulations carried out in Gudmundsson et al. (2023) suggest that the remaining buttressing strength of Thwaites Glacier's ice shelf is rather marginal. The same study finds that Thwaites Glacier's neighboring Smith and Kohler glaciers are, however, substantially buttressed by the downstream Crosson and Dotson ice shelves. Observations show that the thinning of these ice shelves leads to pinning-point loss and that the upstream glaciers are accelerating and retreating (Rignot et al., 2014; Milillo et al., 2022; Miles and Bingham, 2024).

The above mentioned West Antarctic glaciers are principally prone to the so-called marine ice-sheet instability (MISI; Mercer, 1978; Schoof, 2007; Pattyn, 2018; Sergienko, 2022) as they are resting on bed well below sea level that is generally sloping down in landward direction (retrograde bed slope; Morlighem et al., 2020). An ongoing destabilization of these outlets might lead to a disintegration of the West Antarctic Ice Sheet in the long term with the potential to raise global mean sea level by several meters (Bamber et al., 2009; Feldmann and Levermann, 2015b; Ritz et al., 2015; Reese et al., 2023). Among various

processes, mechanisms and feedbacks potentially affecting ice-sheet (de)stabilization (e.g., Brondex et al., 2017; Pegler, 2018a; Sergienko, 2022; Christie et al., 2023), the influence of pinning points has been suggested to play a decisive role regarding the timing, extent and even possible reversal of MISI-type retreat (Goldberg et al., 2009; Matsuoka et al., 2015; Favier et al., 2016).

Here we take an idealized approach to investigate the response of a MISI-prone outlet glacier system to progressive pinning-point ungrounding and re-grounding in numerical simulations. By altering the elevation of the pinning point that stabilizes the outlet glacier on retrograde bed, we show how the depth of the subglacial bed depression can affect (ir)reversibility of glacier retreat and advance in an ensemble of hysteresis experiments. To explain the qualitatively very different responses of the simulated systems, we provide a simple analysis of the steady-state ice flux balance at the GL based on Schoof (2007), quantifying the non-linear influence of both the bed depth at the GL and the pinning-point buttressing strength on ice discharge across the GL. The numerical model and the experimental design are outlined in Sect. 2. The results are presented in Sect. 3 and discussed in Sect. 4 before the conclusions are drawn in Sect. 5.

## 2 Methods

### 2.1 Numerical model

We use the open-source Parallel Ice Sheet Model (PISM; Bueler and Brown, 2009; Winkelmann et al., 2011), which applies a superposition of the shallow-ice approximation (SIA; Morland, 1987) and the shallow-shelf approximation (SSA; Hutter, 1983) of the Stokes stress balance (Greve and Blatter, 2009). In particular, the SSA allows for stress transmission across the GL and thus accounts for the buttressing effect of ice shelves that are laterally confined (Gudmundsson et al., 2012; Fürst et al., 2016; Reese et al., 2018) and/or in contact with a pinning point (Favier et al., 2012; Matsuoka et al., 2015; Wild et al., 2022). The model applies a linear interpolation of the freely evolving GL and accordingly interpolated basal friction (Feldmann et al., 2014). GL migration has been evaluated in the model intercomparison exercises MISMIP3d (Pattyn et al., 2013; Feldmann et al., 2014) and MISMIP+ (Asay-Davis et al., 2016; Cornford et al., 2020). To improve the approximation of driving stress across the GL, the surface gradient is calculated using centered differences of the ice thickness across the GL (Reese et al., 2020). The simulations described below are carried out on a regular horizontal grid of $2\,\mathrm{km}$ resolution, which has been shown to be suitable for accurately modeling dynamics of fast-flowing ice in a previous study (Feldmann and Levermann, 2023).

### 2.2 Setup and experimental design

Our topographic setup is designed to model a MISI-prone outlet glacier that is stabilized by a pinning point in the center of the glacier's ice shelf (Fig. 1). For this purpose we prescribe a modified version of the channel-type MISMIP+ bed topography, $B_{\mathrm{CH}}(x,y)$, as used in Feldmann and Levermann (2023, Appendix C). We superpose a submarine topographic high, $B_{\mathrm{TH}}(x,y)$, in the ice-shelf region of the setup such that the resulting bed topography applied in our simulation is given by

$$B(x,y) = B_{\mathrm{CH}}(x,y) + B_{\mathrm{TH}}(x,y). \tag{1}$$

The geometry of the Gaussian-shaped topographic high is adopted from Favier and Pattyn (2015) and has the form

$$B_{\mathrm{TH}}(x,y) = H_{\mathrm{TH}} \cdot \exp\left(\frac{-(x-x_{\mathrm{TH}})^2 - (y-y_{\mathrm{TH}})^2}{2 \cdot (15\ \mathrm{km})^2}\right). \tag{2}$$

For simplicity, we prescribe the same variance of 15 km (controlling the width of the Gaussian bump) as in Favier and Pattyn (2015). In Eq. (2), $H_{\mathrm{TH}}$ denotes the height of the topographic high relative to the local elevation of the bed trough, $B_{\mathrm{CH}}(x_{\mathrm{TH}}, y_{\mathrm{TH}})$, i.e.,

$$H_{\mathrm{TH}} = -D_{\mathrm{TH}} - B_{\mathrm{CH}}(x_{\mathrm{TH}}, y_{\mathrm{TH}}), \tag{3}$$

where $D_{\mathrm{TH}}$ is the depth of the tip of the topographic high below sea level (Fig. 1b), which is varied in 20-m steps in the course of the hysteresis experiments. We use three versions of $B_{\mathrm{CH}}(x,y)$ which differ in the depth of the bed depression $D_{\mathrm{BD}}$ below sea level (Fig. 1b). These three versions are termed "shallow" ($D_{\mathrm{BD}} = 675$ m), "intermediate" ($D_{\mathrm{BD}} = 750$ m) and "deep" ($D_{\mathrm{BD}} = 900$ m) in the following (Table 1). For each of these three topographies we place the topographic high at three different distances ($x_{\mathrm{TH}} = 500$, 600 and 650 km) away from the ice divide, while in the $y$-dimension the topographic high is

always located at the centerline of the setup ($y_{\mathrm{TH}} = 0$; Fig. 1a). This makes a total of nine distinct hysteresis experiments. Note that the shape of the bed topography is qualitatively the same in all experiments (Fig. 1b): along the setup centerline, the bed elevation drops from an inland sill into an overdeepening (deepest point of the bed depression at $x = 200$ km) and increases toward a coastal sill further upstream (highest point at $x = 500$ km) on which the topographic high is located.

For each hysteresis experiment, consisting of a series of steady-state simulations, the ice-sheet-shelf system is first spun up,
starting from a domain-wide layer of ice of 2000 m thickness. In this initial experiment the elevation of the topographic high is chosen such that the central GL position of the simulated outlet glacier stabilizes 1) on the coastal sill (around $x = 500$ km), i.e., close to the onset of the retrograde slope section (Fig. 2) and 2) upstream of the topographic high such that an ice-shelf pinning point emerges that is clearly distinct from the main GL of the outlet (except for the experiments in which the topographic high is located on the coastal sill; see Figs. 2a,d,g). Note that for one hysteresis experiment (topographic parameters: $D_{\mathrm{BD}} = 900$ m
and $x_{\mathrm{TH}} = 600$ km; see Fig. 3c) the initial topographic-high elevation is chosen higher than would be required by the above two criteria in order to obtain a retreat branch that matches the extent of the long re-growth branch of this hysteresis experiment. The chosen initial elevations of the topographic high range from $-20$ m to $-200$ m throughout the nine hysteresis experiments. Each resulting initial state of the ice-sheet-shelf system is then perturbed by a 20-m lowering of the topographic-high elevation. The simulation is run until the glacier has reached a new equilibrium, i.e., until changes in the glacier volume have become
negligible (no trend in total ice volume, with remaining fluctuations $< 1‰$) and the GL is in steady state again (Fig. S1). Starting from the new equilibrium, again the topographic-high elevation is lowered by 20 m and the system is run into steady state. These perturbation experiments are carried out repeatedly until the outlet's ice shelf has detached from the topographic high. The sign of the perturbation is then reversed, i.e., the unpinned equilibrium state is now perturbed by a 20-m increase of the topographic-high elevation. The step-wise increase in topographic-high elevation after each equilibration is continued until
the original topographic-high elevation is reached. The step-wise forcing and the time evolution of the response of the glacier's

centerline GL position throughout a complete series of subsequent steady-state experiments is exemplarily shown in Fig. S1 for each of the three different bed-depression depth. The incremental change in topographic-high elevation realizes a gradual reduction or increase in pinning-point induced ice-shelf buttressing in a synthetic but minimal-invasive manner. That is, the perturbation alters the ice-shelf buttressing solely through the change in contact area between the ice shelf and the topographic high, without directly affecting the thickness, extent or softness of the ice shelf. A more detailed reasoning for applying this perturbation type and a discussion of its benefits and drawbacks are given in Sect. 4. The experiments described above vary the location and the elevation of the topographic high, as well as the *depth* of the bed depression. The role of the *length* of the bed depression is examined in a separate set of simulations, presented in the Appendix.

## 2.3 Forcing and boundary conditions

All experiments prescribe the same surface accumulation rate, which is constant in space and time (see Table 1, also for more parameters). The same applies for the ice softness, thus the simulations are *not* thermomechanically coupled. Basal melting is set to zero. Ice is cut off from the ice shelf and thus calved into the ocean beyond a fixed position, $x_{CF} = 780$ km. Note that we avoid prescribing a boundary condition at $x = 0$ by mirroring the setup (including the bed perturbations) along this axis, producing an ice sheet that consists of two symmetric outlet glaciers that share the same ice divide (as we have done in previous studies, e.g., Cornford et al., 2020; Feldmann et al., 2022; Feldmann and Levermann, 2023). The entire computational domain thus spans from $-800$ km to $+800$ km in the $x$ direction. Due to the symmetry of the system with respect to the ice divide, we here only consider one of the two outlet glaciers, i.e., the right-hand half of the computational domain ($x \geq 0$). Regarding the lateral margins of the computational domain, i.e., $y = \pm160$ km, periodic boundary conditions apply. Basal friction is calculated according to a Weertman-type power law (Cornford et al., 2020, Eq. 3). We will refer to the experiments described above as laterally confined simulations throughout the remainder of the text, the results of which are presented in Sect. 3.1. Based on these experiments we conduct a small set of additional simulations for which we removed the lateral confinement of the prescribed bed topography. These laterally unconfined simulations are detailed in Sect. 3.2.

## 3 Results

### 3.1 Laterally confined simulations

The general response of the simulated outlet glaciers to each incremental topographic-high lowering is characterized by a retreat of the GL due to the reduced buttressing effect of the pinning point (red curves in Fig. 3). Conversely, the subsequent step-wise rise in elevation of the topographic high leads to GL re-advance due to the increase in pinning-point buttressing (blue curves in Fig. 3). Regarding the complete cycle of topographic-high lowering and subsequent raising we identify three qualitatively different types of glacier response:

1. **Reversible behavior:** regardless of whether the glacier is retreating or re-growing, for a given topographic-high elevation the steady-state GL position is approximately the same. In the hysteresis diagram the path of the steady-state positions

of the re-advancing GL is approximately the same as the path of the previously retreating GL (compare blue and red curves in Fig. 3a).

2. **Hysteretic behavior:** the re-growing glacier reverts to its original (initial) state but the steady-state GL positions depend on the history of glacier evolution. In other words, after large-scale retreat, the GL of the re-growing glacier requires a higher topographic-high elevation (more pinning-point buttressing) to re-advance towards its original position. This hysteresis gap between the path of GL retreat and the path of GL re-advance, $\Delta D_{\text{TH}}$, is highlighted in Fig. 3b.

3. **Lock-in behavior:** the re-growing glacier does not revert to its original state. After ice-shelf re-grounding on the topographic high the GL advances only marginally and remains situated on the inland sill near the ice divide. The glacier thus remains locked into a state that is similar to the fully retreated state after ice-shelf ungrounding (Fig. 3c).

Out of the nine sets of hysteresis experiments, three yield reversible GL evolution (marked "R" in Fig. 4), four a hysteretic behavior ("H") and two show a lock-in effect ("L"). The type of response depends on the bed-topographic configuration, i.e., on the combination of the size of the bed depression and the location of the topographic high (Fig. 1b). As can be seen from Fig. 4, a deeper bed depression and a topographic high that has a larger distance from the ice divide favor hysteretic and lock-in behavior.

In each series of experiments the retreating GL eventually passes the deepest point of the bed depression, finding a final stable position on the prograde bed section upstream of the bed depression (Figs. 1b and 2). A flat bed depression and/or a topographic-high location less distant from the ice divide lead to more gradual steps of GL retreat between two subsequent experiments (on the order of 10 km; see panel a in Figs. 3 and S1-3). In contrast, a deep bed depression and a more distant topographic-high location foster an abrupt change in GL position ($\gtrsim 200$ km) once the topographic-high depth crosses a certain threshold, resulting in a steep hysteresis curve (red curves in panels b and c of Figs. 3 and S1-S4). Thus, in these experiments the simulated glacier transitions from a relatively large, advanced state (GL on coastal sill) into a small, retreated state (GL on inland sill, Fig. 2). During this MISI-type retreat the ice shelf remains pinned on the topographic high. Further lowering of the topographic high then induces comparatively small, gradual GL retreat on the prograde slope of the inland sill until the ice shelf ungrounds from the topographic high. The topographic-high depth at which the ice shelf ungrounds, $D_{\text{TH,ungr}}$, increases with decreasing distance of the topographic high from the ice divide ($D_{\text{TH,ungr}} = 240$ m for $x_{\text{TH}} = 650$ km, $D_{\text{TH,ungr}} = 280$ m for $x_{\text{TH}} = 600$ km and $D_{\text{TH,ungr}} = 400$ m for $x_{\text{TH}} = 500$ km): a topographic high located closer to the ice divide acts on a thicker part of the ice shelf and thus requires a stronger lowering in order to achieve the detachment of the ice shelf (Fig. 2).

Reversal of the perturbation, i.e., rising instead of lowering the elevation of the topographic high, leads to the re-grounding of the ice shelf on the topographic high already at the first step. The re-emerged pinning point exerts a buttressing force on the upstream ice again. In the further course of the step-wise rise in topographic-high elevation the contact area between the bed and the ice shelf increases, strengthening the pinning-point buttressing and thus causing a GL re-advance into the bed depression. A deeper bed depression and a more distant topographic high hamper the GL in passing the depression (Figs. 3

and S1-S3). Consequently, larger $D_{\mathrm{BD}}$ and/or $x_{\mathrm{TH}}$ increase the gap between the two curves in the hysteresis diagram (denoted by $\Delta D_{\mathrm{TH}}$ in Figs. 3b and 4). This gap measures the amount of additional change in the perturbation, required to tip back the glacier from its retreated to its advanced state on the path of re-growth (blue curves in Figs. 3b, S2b,c and S3b) compared to the path of retreat. In other words, $\Delta D_{\mathrm{TH}}$ is the additional increase of the topographic-high elevation required to induce GL re-advance from the bed depression onto the coastal sill, compared to the reversible case (corresponding to $\Delta D_{\mathrm{TH}} = 0$). If the depth of the bed depression and the distance of the topographic high are sufficiently large, the GL does not pass the depression in downstream direction in the course of the elevation increase of the topographic high and the glacier remains in a small, retreated state (Figs. 2h and i).

## 3.2 Laterally unconfined simulations

For more insight into why the glacier response behavior is affected by the depth of the bed depression and the distance of the pinning point, we conduct a small set of additional experiments to examine the glacier's flux balance under flowline conditions. For this purpose we have removed the lateral confinement of the prescribed bed topography by doubling the bed-trough width and cutting off the computational domain at $y = \pm 80\,\mathrm{km}$ (compare Figs. S6a to S7a), referring to these modified simulations as laterally unconfined in the following. In the flow band around the centerline of the computational domain the bed profiles of the laterally unconfined setup and the laterally confined setup are the same. Since in the laterally unconfined case the simulated ice stream is not restrained laterally, its flow is generally in $x$ direction and the $y$ component of the velocity field is non-zero only in the vicinity of the ice-shelf pinning point (compare Figs. S6b to S7b). We make use of these flowline conditions that prevail far enough upstream of the pinning point in a flux balance analysis that is presented in Sect. 3.2.1.

Note that in the laterally unconfined simulations the ice-shelf pinning on the topographic high is the only source of buttressing compared to the laterally confined experiments in which the central pinning-point buttressing is complemented by a backforce generated at the lateral shear margins of the ice shelf. The reduced backstress on the upstream grounded ice implies a larger ice discharge across the GL (outflux) with major implications for the steady states and the glacier response in the laterally unconfined case. First, without any modifications, the model initialization would result in a much smaller outlet compared to the laterally confined case. To compensate for this effect we prescribe a substantially higher surface accumulation rate ($a = 1\,\mathrm{ma}^{-1}$ instead of $0.15\,\mathrm{ma}^{-1}$) in the laterally unconfined setup, obtaining an initial steady-state glacier profile that is qualitatively comparable to the laterally confined case (compare Figs. 2 to S5). For consistency, the other model parameters are left unchanged. Second, the step-wise reduction in pinning-point buttressing eventually leads to the complete disintegration of the grounded ice sheet even before the ice shelf ungrounds from the topographic high. Third, the re-advance of the GL in response to raising the topographic high is strongly limited compared to the laterally confined case. In fact, in the absence of confinement-induced buttressing, pinning-point buttressing is insufficient to cause glacier re-advance toward its original state even for the case of a flat bed depression and a much higher topographic-high elevation than the original value (Figs. S4 and S5).

### 3.2.1 Flowline flux balance analysis

In the laterally unconfined case, the velocity of the ice-sheet-shelf system is zero in the $y$-dimension for the region upstream of $x = 400$ km (see straight velocity contours in Fig. S7b), where it thus can be treated as a flowline ice stream in $x$ direction. In the retreated state, i.e., when the grounded part of the glacier has retreated onto the inland sill and is thus located far upstream of the pinning point, the GL is straight and parallel to the velocity contours. For such a buttressed glacier in flowline, the steady-state ice discharge across the GL, i.e., the outflux $Q_o(x_{\mathrm{GL}})$, can be calculated analytically according to the boundary layer theory by Schoof (2007, Eq. 29),

$$Q_o(x_{\mathrm{GL}}) = -c\,\Theta(x_{\mathrm{GL}})^{\frac{n}{m+1}}\,B(x_{\mathrm{GL}})^{\frac{m+n+3}{m+1}}, \tag{4}$$

as a function of the bed elevation, $B(x_{\mathrm{GL}}) = -\frac{\rho_i}{\rho_o}h(x_{\mathrm{GL}})$, with ice thickness $h(x_{\mathrm{GL}})$, and of the buttressing ratio $\Theta(x_{\mathrm{GL}})$ at the GL position, $x_{\mathrm{GL}}$, respectively (Schoof, 2007; Gudmundsson et al., 2023). The parameter $n$ denotes the exponent in Glen's flow law (Glen, 1955) and $m$ is the exponent in the Weertman friction law (Cornford et al., 2020, Eq. 3). The factor $c$ is constant in space and time,

$$c = \left[ \frac{A(\rho_i g)^{n+1}(1 - \rho_i/\rho_o)^n}{4^n C} \left( \frac{\rho_o}{\rho_i} \right)^{m+n+3} \right]^{\frac{1}{m+1}}, \tag{5}$$

in which $A$ is the ice softness, $C$ is the basal-friction coefficient, $g$ denotes the gravitational acceleration and $\rho_i$ and $\rho_o$, are the ice and ocean densities, respectively (values for all parameters given in Table 1). Note that, as we are considering marine outlet glaciers, $B(x_{\mathrm{GL}})$ is always negative and since $c > 0$ the outflux $Q_o(x_{\mathrm{GL}})$ is positive.

For the parameter values used in our experiments, Eq. (4) reads

$$Q_o(x_{\mathrm{GL}}) = -c\,\Theta(x_{\mathrm{GL}})^{\frac{9}{4}}\,B(x_{\mathrm{GL}})^{\frac{19}{4}}. \tag{6}$$

The buttressing ratio, $\Theta(x_{\mathrm{GL}})$, relates the ice-internal stresses at the GL to the stresses exerted at the calving front due to ocean pressure. If $\Theta(x_{\mathrm{GL}}) < 1$, the stresses at the GL are smaller compared to the ocean pressure and the ice shelf provides buttressing, reducing the GL flux. If $\Theta(x_{\mathrm{GL}}) > 1$, the opposite is the case, i.e., traction at the GL is larger than the ocean pressure, which means that the ice shelf is "pulling" which increases the ice discharge across the GL (Gudmundsson, 2013). In case of $\Theta(x_{\mathrm{GL}}) = 1$, the presence of the ice shelf has no effect on the upstream stresses, i.e., the stresses at the GL are equal to the ocean pressure at the calving front. For instance, such conditions prevail in the absence of pinning points for an unconfined ice shelf in the flowline case (only one horizontal dimension), such that the ice shelf does not provide any buttressing nor "pulls" the flow at the GL. In this case, the GL flux depends solely on the bed topography and thus Eq. (6) reduces to

$$Q_o(x_{\mathrm{GL}}) = -c\,B(x_{\mathrm{GL}})^{\frac{19}{4}}. \tag{7}$$

This function is visualized by the grey curves in Figs. 5e and f for the centerline profile of the bed depression, demonstrating how the bed profile is mirrored by the GL flux in a highly non-linear way. In case of a deep bed depression the peak of the flux

curve is much larger than in the shallow case (flux larger by about $400\ \%$ for a $33\ \%$ deeper depression). Note that this curve gives the steady-state ice discharge for any potential GL position along the bed profile. For an outlet glacier in equilibrium, this outflux $Q_o$ has to be balanced by an influx $Q_i$ which in flowline is given by the integral of the surface accumulation rate $a$ between the ice divide ($x = 0$) and the GL position, i.e.,

$$Q_i = a\,x_{\mathrm{GL}}. \tag{8}$$

The slope of this linear function in $x_{\mathrm{GL}}$ is determined by the surface accumulation rate (black curves in Figs. 5e and f). For the parameters and bed topographies prescribed in our experiemnts, the curves of in- and outflux do not intersect in the absence of buttressing, and thus a steady-state outlet glacier cannot exist in this configuration according to the analytical calculations. This is in agreement with our laterally unconfined simulations in which the glacier retreats all the way to the ice divide and thus vanishes in the course of the step-wise topographic-high lowering (and thus buttressing reduction) for a pinning-point depth of
$D_{\mathrm{TH}} = 300$ m even before its ice shelf ungrounds from the topographic high (Fig. S5). At this stage, an ice shelf remains that covers the entire computational domain and is pinned on the topographic high.

     Reverting the perturbation sequence at this stage, i.e., reducing the topographic-high depth to $D_{\mathrm{TH}} = 280$ m, leads to buttressing-induced ice shelf thickening that is sufficient to cause the re-grounding of the ice shelf on the inland sill such that a small glacier re-emerges ($x_{\mathrm{GL}} > 0$; Fig. S4). Now the curves of in- and outflux have an intersection (Figs. 5e and f),
predicting a stable GL position since $Q_o$ increases faster than $Q_i$ at the GL (Schoof, 2012). In the following, we focus on the subsequent glacier re-advance in response to the step-wise increase in the topographic-high elevation. The backstress exerted by the ice-shelf pinning point on the upstream ice is associated with values of $\Theta < 1$. This can be seen from the steady-state flowline profiles of $\Theta$ which we infer from the simulations, based on the traction normal to the GL along the centerline of the setup (Figs. 5c and d). Note that in a strict sense, $\Theta$ may only be evaluated at the GL position to calculate the GL flux $Q_o$
corresponding to the steady state. Thus, the resulting curves from calculating $Q_o$ for the profile of $\Theta$ show the theoretical GL flux under the assumption that the $\Theta$ profile would not change for a change in the GL position. That is, the gap between the $Q_o$ profiles and $Q_i$ at a location $x$ downstream of the GL can be interpreted as the additional amount of flux reduction that would be required to advance the GL to this location but is not provided under the buttressing conditions of the current steady state.

     In the vicinity of the GL, the diagnosed $\Theta$ values range roughly between 0.4 and 0.65 with the lower values corresponding
to higher topographic-high elevations, i.e., stronger pinning-point buttressing. As can be seen from the colored curves in Figs. 5e and f, the pinning-point buttressing leads to a flattening (vertical compression) of the GL flux profile compared to the unbuttressed case (compare to grey curves associated with $\Theta = 1$). In fact, this reduction in GL flux allows for stable solutions of the GL position as can be seen from the intersects between in- and outflux curves and the simulated glacier profiles. With increasing pinning-point buttressing, i.e., increasing topographic-high elevation, the flattening of the curves becomes more
pronounced (Figs. 5e and f). In case of a shallow depression, the gap between the curves of in- and outflux becomes very small for the entire overdeepening (Fig. 5e). These conditions ease the advance of the GL within the region of the bed depression. The situation is different in the deep-depression case, in which, despite the pinning-point induced reduction in outflux, a substantial

gap between the curves of in- and outflux remains along the central part of the overdeepening (outflux about one order of magnitude larger than influx at the deepest point of the bed depression; see Fig. 5f). With similar buttressing values compared

to the flat-depression case, the compensation of the much larger GL flux due to the deeper bed depression remains much more limited, resulting in diminished GL advance.

In the laterally unconfined setup, the buttressing effect of the pinning point alone is not strong enough to induce GL advance past the deepest point of the bed depression. Thus, for the parameter configuration and the range of topographic-high elevations considered in this study, the presence of confinement-induced buttressing is a requirement to avoid lock-in of the retreated

glacier but to allow for its recovery. This holds true even for the case of a shallow bed depression (compare Figs. 3a to S4a). The role of confinement-induced buttressing is further explored in the Appendix.

## 4    Discussion

Our simulations demonstrate how an idealized MISI-prone outlet glacier that is stabilized on a retrograde bed by pinning-point induced buttressing can undergo large-scale and irreversible retreat in response to a step-wise buttressing reduction. The

prescribed bed topography plays a key role in determining (ir-)reversibility of glacier retreat and re-advance. Our topographic setup (Fig. 1) is based on previous studies (Gudmundsson et al., 2012; Cornford et al., 2020) that simulated a stable GL position on retrograde bed in the absence of pinning-point buttressing. In those experiments buttressing was provided exclusively due to the relatively narrow to moderate bed trough confining the streaming trunk of the glacier laterally (prescribed confinement widths of roughly 50 to 100 km). Here we use a much wider confinement width of 160 km which is similar to the observed

width of Thwaites Glacier's fast flowing trunk at the GL (e.g., Davison et al., 2023, Fig. 1a). In the absence of a pinning point, such a confinement would be too wide to prevent glacier collapse during model initialization, as we have shown in earlier simulations (Feldmann and Levermann, 2023). This is consistent with results from Goldberg et al. (2009) who also found that ice-shelf grounding on a central topographic high can prevent MISI-type retreat and concluded that the presence of such a pinning point can have a similar effect as halving the ice-shelf width (and thus increasing the buttressing effect based on

parameterized lateral drag at the ice-shelf margins). In the present study, we find that the GL destabilizes once the topographic-high elevation is lowered beyond a critical threshold but only if the glacier rests on a sufficiently deep bed depression (Figs. 3, and S1-S3). GL stabilization on the retrograde slope takes place for topographic-high elevations between $-380$ and $-40$ m. A steeper retrograde slope and a larger distance of the pinning point from the ice divide require higher topographic-high elevations (associated with a stronger buttressing effect of the pinning point) to stabilize the GL on the retrograde bed. Antarctic-wide

mean bed elevations beneath ice rises and ice rumples, calculated from present-day observations, range from about $-200$ to $-300$ m (Matsuoka et al., 2015). Thus, particularly the initial topographic-high depths prescribed in our simulations are very shallow compared to the observations.

In our laterally confined simulations, that comprise both horizontal dimensions, ice-shelf buttressing emerges as a combination of 1) horizontal shearing within the ice at the lateral, topographically confined ice-stream margins (confinement-induced

buttressing) and 2) basal drag where the ice shelf is pinned on the topographic high (pinning-point induced buttressing). In the laterally unconfined simulations the pinning point is the only source of buttressing. The comparison to the laterally confined simulations reveals that the additional buttressing effect of a laterally confined ice shelf facilitates glacier re-advance substantially since in the laterally unconfined experiments the glacier remains locked in its retreated state even for the flattest bed depression and a substantially higher surface accumulation rate (Figs. 5, S4 and S5). The 500-m deep and 160-km wide topographic confinement, prescribed in the laterally confined simulations, strongly confines the fast glacier (ice-stream) flow throughout the entire length of the computational domain. In contrast to that, the observed fast flowing trunk of Thwaites Glacier is much less confined by the subglacial bed topography and its width increases into the interior, doubling to about 300 km over the first 200 km upstream of the present-day GL (Davison et al., 2023).

The laterally unconfined simulations serve as a quasi-flowline setting to which we can apply a flux balance analysis according to Schoof (2007), involving the strength of (unparameterized) pinning-point buttressing, $\Theta$, and the bed depth at the GL (Eq. 6). Since Schoof (2007), other analytic frameworks have been introduced, designed for specific investigations of the influence of individual possible regulators of the GL flux of marine buttressed ice sheets, such as basal ice-shelf melting, calving or (parameterized) lateral drag (e.g., Schoof et al., 2017; Haseloff and Sergienko, 2018; Pegler, 2018b). In the mentioned frameworks, the buttressing factor $\Theta$ (Eq. 6) is an explicit function of, e.g., the calving-front thickness, the ice-shelf width and length, or the prescribed lateral shear-stress magnitude. Here we stick to the more general formulation by Schoof (2007) and diagnose $\Theta$ directly from our simulations. In particular, this allows us to demonstrate how the pinning-point buttressing increasingly suppresses the GL flux in response to the lifting of the topographic high.

A characteristic outcome of our experiments is that outlets which rest on a deeper bed depression and which are less buttressed, generally show a greater susceptibility to abrupt and irreversible (MISI-type) retreat (Figs. 3, 4 and S1-S3). Applying our results to the real world in a qualitative sense, we thus expect that wider outlet glaciers with weak lateral confinement, like Thwaites Glacier, would be more susceptible to abrupt retreat and more likely exhibit hysteretic or lock-in behavior than narrower outlets with a strong topographic confinement like Pine Island Glacier (MacGregor et al., 2013; Rignot et al., 2014; Morlighem et al., 2020; Schwans et al., 2023). We stress that our simulations are not able to represent a specific glacier but are rather supposed to relate large-scale outlet glacier retreat (and possible re-advance) to characteristic large-scale features of the bed geometry in a general sense. The many differences between our idealized simulations and nature limit particularly the quantitative applicability of our results to the real world. In the following we give a reasoning for our particular choice of perturbation and discuss several simplifying aspects of our simulations and how they affect our results.

The perturbations applied in our simulations are of synthetic character and would not be observed in nature. In the real world, changes in pinning-point buttressing would result from, e.g., ice-shelf thickness changes (due to sub-ice-shelf melting/refreezing; Reese et al., 2018; Gudmundsson et al., 2019), ice-shelf weakening/fracturing in the vicinity of the pinning point (Benn et al., 2022; Sun and Gudmundsson, 2023; Surawy-Stepney et al., 2023), iceberg calving (Arndt et al., 2018) or glacial isostatic adjustment (GIA; Adhikari et al., 2014; Matsuoka et al., 2015). Thus, our approach of directly manipulating the topographic-

high elevation, is simplistic and unrealistic. At the same time this approach reduces the complexity of the effects resulting from the perturbation itself. In fact, we here strive to apply a perturbation that provides a good control over the buttressing effect of the pinning point (mediated through the contact area between the topographic high and the ice shelf) but minimizes any other direct interference of the perturbation with the rest of the system. For instance, applying basal ice-shelf melt in our simulations would indeed decrease the ice shelf's contact area with the topographic high and thus reduce pinning-point buttressing. However, at the same it would directly alter the overall ice-shelf geometry and thus directly affect the strength of the confinement-induced buttressing, which we want to avoid. Introducing fractures to the ice shelf to regulate pinning-point buttressing would also directly affect the dynamics of the simulated system due to the non-locality of the underlying stress balance.

Our simulations neglect basal ice-shelf melting which is an important driver of Antarctic ice-sheet dynamics, shaping the geometry of ice shelves, often reducing their buttressing effect and thus increasing the discharge of the outlet glaciers around Antarctica (Dutrieux et al., 2013; Reese et al., 2018; Gudmundsson et al., 2019). The inclusion of basal melting in our simulations would lead to an overall thinner ice shelf, facilitating ice-shelf unpinning (GL retreat) and hampering re-grounding (GL advance). This might expand the regimes of hysteretic and lock-in behavior at the expense of the reversibility regime in our parameter space. In fact, due to the incremental thickening of the ice shelf with increasing topographic-high elevation, even in the simulated lock-in cases the glacier is expected to eventually re-advance to its original state due to thickening-induced ice-shelf grounding on the upstream side of the coastal sill (note the proximity of the ice-shelf draft to the sill in Figs. 2h and i). Since the ice-shelf thickness is approximately uniform across the entire bed trough (except for the vicinity of the topographic high), the grounding of the ice shelf upstream of the coastal sill would cover the entire bed-trough width and thus lead to a closing of the ice-shelf cavity. That is, a water volume on the order of $1000 \text{ km}^3$ ($10^{15}$ L) would be trapped within the bed depression, confined by the ice-shelf draft at the top, the overdeepened bed at the bottom and the lateral walls of the bed trough.

Prescribing a fixed bed topography in each simulation, we neglect possible GIA effects (e.g., Adhikari et al., 2014; Matsuoka et al., 2015; Barletta et al., 2018). Including GIA in our experiments would account in particular for variations in pinning-point buttressing that originate inherently from the evolution of the coupled glacier-bed system and thus strongly limit our possibilities to regulate pinning-point buttressing. Furthermore, it would likely introduce several feedback mechanisms (Albrecht et al., 2023), the analysis of which is beyond the scope of this study. For instance, a rebound of the bed induced by glacier retreat would reduce the depth of the bed depression and thus facilitate glacier re-advance. It could thus counteract the lock-in effect found for the deepest bed depression used in our simulations and shift the lock-in regime into regions of greater bed-depression depth. Note that the bed depressions (and thus the bed slopes) chosen here are rather less pronounced compared to observations in the Amundsen Sea Embayment. For instance, the bed along a central transect through Thwaites Glacier drops by about $-750 \text{ m}$ over the first $250 \text{ km}$ upstream of the GL, resulting in an average bed slope of $s \approx -3 \cdot 10^{-3}$ (Morlighem et al., 2020; Sergienko and Wingham, 2022). In our simulations the steepest part of the retrograde slope has values of $s_{D_{\text{BD}}=675} \approx -3.75 \cdot 10^{-4}$, $s_{D_{\text{BD}}=750} \approx -7.5 \cdot 10^{-4}$, $s_{D_{\text{BD}}=900} \approx -1.5 \cdot 10^{-3}$, for the three different depths of the overdeepening, respectively. In our additional experiments that prescribe a shortened bed depression (see Appendix) the retrograde

slope is steeper by a factor of 3 (Fig. A2). For the deepest version of the shortened bed depression this results in an average retrograde bed slope that is comparable to the observed mean slope along the Thwaites transect mentioned above. However, the bulk of our simulations represents much shallower retrograde slopes (average slopes ranging from $-2.5 \cdot 10^{-4}$ to $-1 \cdot 10^{-3}$),
e.g., resembling the more gentle bed slopes of the continental shelves below Antarctica's Ross and Filchner-Ronne ice shelves (Kingslake et al., 2018, see their Extended Data Fig. 3). It has been suggested that past GL migration after the Last Glacial Maximum in these regions has been substantially affected by GIA, as bed rebound and associated ice-shelf grounding may have partly reverted climate-induced GL retreat (Kingslake et al., 2018; Wearing and Kingslake, 2019). Evidence provided in these studies suggests that the formation of Crary Ice Rise and Henry Ice Rise due to the uplift of topographic highs in
response to large-scale ice-sheet retreat led to the regional re-advance of the GLs of Ross Ice Shelf and Filchner Ice Shelf, respectively. Our prescribed synthetic increase of the topographic-high elevation in the course of the re-growth branch of our hysteresis experiments mimmicks this pinning-point uplift to some extent. We chose the magnitude of the applied step-changes of the topographic-high elevation as compromise between the graduality of pinning-point buttressing change and the number of numerical simulations, i.e., the computational cost. Smaller perturbation steps would in particular allow for a more accurate
quantification of the critical thresholds of abrupt GL retreat and re-advance, but leave the conclusions of our study unchanged.

Variations in basal friction (between ice and bed) are not investigated in our study. A comparison between Ross Ice Shelf's various pinning points concluded that the magnitude of their effective resistance to the ice flow does not only depend on the area and elevation of the pinning point but also on the nature of the subglacial material on which the ice shelf is grounded (Still et al., 2019). Idealized simulations showed that a higher degree of friction at the ice-bed interface favors the emergence of pinning
points of larger horizontal extent which provide more buttressing to the upstream flow (Henry et al., 2022). Consequently, we expect that a stronger basal resistance would favor reversibility of retreat and advance.

Since we apply a fixed calving-front position in our simulations, GL retreat directly translates into a length increase of the confined ice shelf and thus increases its buttressing strength, a finding which has been reported before by Sergienko and Haseloff (2023) for a flowline setting. In our experiments, this effect facilitates GL re-advance in the case of prior stronger
GL retreat, as shown by a comparison between simulations that use the default bed-depression length versus a shortened version of the bed depression (Appendix, compare Figs. 3 to A3). This highlights the importance of the choice of the calving law or calving condition when modeling the evolution of buttressed marine ice sheets, since it can have profound influence on the magnitude of potential GL migration and particularly on its (ir)reversibility (see also Sergienko and Haseloff, 2023). For instance, simulations of the future long-term evolution of the (West) Antarctic Ice Sheet often prescribe fixed present-
405 day calving fronts (e.g., Joughin et al., 2014; Feldmann and Levermann, 2015a; Reese et al., 2023), thus implying the above mentioned positive contribution of GL retreat to ice-shelf buttressing. Allowing for a dynamic calving-front evolution instead would likely alter the timing, extent and reversibility regime of the large-scale, unstable retreat of the West Antarctic Ice Sheet found in these studies.

## 5  Conclusions

Our idealized simulations highlight that the stability and (ir-)reversible migration of MISI-prone outlet glaciers can be strongly affected by the interplay of 1) variations in pinning-point buttressing and 2) the depth of their subglacial bed depression. Abrupt, unstable GL retreat (MISI) occurs for sufficiently steep retrograde slopes once a critical perturbation threshold (elevation of the topographic high that controls the buttressing strength of the pinning point) is crossed (Figs. 3 and S1-S3). This retreat is irreversible, meaning that glacier re-advance under reversal of the perturbation is either of hysteretic nature or so marginal that

the glacier remains locked in a small, retreated state (Figs. 2 and 4). Conversely, in the case of a comparatively flat retrograde slope, glacier retreat and re-advance follow the perturbation sequence gradually and are reversible. However, in the absence of the strong lateral topographic confinement (substantially diminished overall ice-shelf buttressing), the glacier does not recover from its retreated state even for a shallow bed depression (Figs. 5, S4 and S5).

While the application of our results to the real world is limited in a quantitative sense due to the idealized nature of our

simulations, our findings qualitatively suggest that outlets that rest on a deep bed depression and are weakly buttressed, such as Thwaites Glacier, are more susceptible to abrupt and irreversible (MISI-type) retreat than stronger buttressed glaciers on more moderate retrograde slope, such as Pine Island Glacier. On a larger scale, our results suggest that West Antarctica's bed-topographic configuration with its very wide marine basin that drops into one of the deepest regions of the Antarctic continent does not only provide favorable conditions for unstable retreat but also would strongly hamper possible glacier re-

advance after a potential large-scale retreat. According to our findings, the (ir)reversibility of large-scale GL retreat may also be strongly affected by calving-front migration and associated changes in ice-shelf buttressing.

*Code and data availability.*  The model code used in this study is based on PISM stable version 1.0 and can be obtained from https://doi.org/10.5281/zenodo.6531439 (Feldmann, 2022). Model output and scripts to reproduce the simulations and figures of this paper can be found at https://doi.org/10.5281/zenodo.12793017 (Feldmann, 2024).

**Appendix A:  Influence of length of bed depression**

The laterally confined simulations presented in the main text investigate the influence of the position and the elevation of the topographic high (the pinning point), and the *depth* of the bed depression, on glacier retreat and re-advance. Here we analyze additional results from experiments with an shortened *length* of the bed depression. For this purpose, the inland sill (prograde bed section; see Fig. 1) is shifted $200\,\mathrm{km}$ in downstream direction (Fig. A1). This shift roughly halves the bed-depression

length, shortening the retrograde slope section from $300$ to $100\,\mathrm{km}$ and thus steepening the retrograde slope by a factor of 3. For simplicity, the bed elevation between the ice divide ($x = 0$) and the shifted upstream end of the prograde bed section (now at $x = 200\,\mathrm{km}$) is assumed to be a flat plateau of $-400\,\mathrm{m}$ elevation (Fig. A1). The resulting modified bed geometry preserves the geometry of the prograde bed section on which the glacier eventually stabilizes in the course of the step-wise pinning-point

buttressing reduction. We conduct three hysteresis experiments for the same three bed-depression depths that were used in the main simulations, following the step-wise perturbation procedure described in the Methods Section. The location of the topographic high, $x_{\mathrm{TH}} = 600$ km, is the same in all three experiments.

In the course of the step-wise lowering of the topographic high (retreat branch of the hysteresis curves), we find that for all three bed-depression depths, unstable retreat (MISI) is triggered once a critical topographic-high depth is crossed (at $D_{\mathrm{BD}} = 160$ m for the deep and intermediate depression, and $D_{\mathrm{BD}} = 180$ m for the shallow depression). Thus, the gradual GL retreat down the retrograde slope found in the original shallow-depression case (Fig. 3a) has turned into large-scale retreat, resulting from the prescribed steeper retrograde slope on which the GL cannot stabilize (Fig. A3a). With increasing bed-depression depth, GL stabilization on the inland sill occurs further upstream which also holds true for the final GL position after the ice shelf has unpinned from the topographic high (at $D_{\mathrm{TH}} = 280$ km in all three hysteresis experiments; Fig. A3). This is qualitatively similar to the results obtained from the original simulations (compare to Fig. 3). What differs is the magnitude of retreat with respect to the deepest point of the bed depression, i.e., the inland end of the retrograde slope section: the retreat is much more pronounced in the case of the shortened bed depression (up to $> 50$ km difference). Due to the larger distance of the GL from the deepest point of the bed depression, stronger pinning-point buttressing is required in order to re-advance the GL towards its initial position after reversal of the perturbation compared to the original simulations. In fact, in the intermediate case, raising the topographic-high to the very shallow depth of $D_{\mathrm{TH}} = 20$ m is not sufficient to induce GL re-advance beyond the bed depression and the glacier remains locked in its retreated state (in contrast to the hysteretic but complete re-advance found in the original simulations; compare Figs. 3b and A3b). In case of a shallow bed depression, the former GL reversibility turns into hysteretic behavior with a hysteresis gap of $\Delta D_{\mathrm{TH}} = 60$ m (compare Figs. 3a and A3a). The deep-depression case shows lock-in behavior for both bed-depression lengths, though for the shortened bed depression the GL is located about twice as far upstream of the deepest point of the bed depression at the final step of the perturbation sequence (compare Figs. 3c and A3c).

It may seem counter-intuitive that a shorter bed depression promotes hysteretic glacier retreat and lock-in compared to a bed depression that is twice as long. The reason for this outcome lies in the stronger buttressing effect of the longer and thus thicker ice shelf resulting from the stronger GL retreat through the long bed depression: since all our experiments prescribe a fixed calving front position, GL retreat directly translates ice-shelf lengthening. Consequently, GL retreat through the longer bed depression results in a longer ice shelf (compare second column of Fig. 2 to Fig. A2). When the ice shelf finally ungrounds from the topographic high, its laterally confined part (spanning from the centerline GL location to approximately $x = 600$ km; see Fig. S6) is about $50\%$ longer compared to the short-depression case. The increase in the length of the ice-shelf confinement is associated with more lateral resistance that leads to thicker ice upstream (van der Veen, 2013). In other words, the stronger confinement-induced buttressing supports a steady-state GL position at greater bed depth on the inland sill. Thus, though in the long-depression case the GL retreats much further inland, the GL stabilizes closer to the deepest point of the bed depression, facilitating glacier re-advance during the step-wise increase in pinning-point buttressing.

*Author contributions.* JF designed the study and performed the numerical simulations. JF analyzed the data and wrote the paper with contributions from AL and RW.

*Competing interests.* The contact author has declared that none of the authors has any competing interests.

*Acknowledgements.* JF would like to thank Ronja Reese for the insightful exchange on ice-shelf buttressing. The authors are very grateful to the reviewers of this paper who provided valuable comments that helped to improve the manuscript. This work was supported by the Deutsche Forschungsgemeinschaft (DFG) in the framework of the priority programme "Antarctic Research with comparative investigations in Arctic ice areas" SPP 1158 through grant WI 4556/6-1. JF and RW are grateful for support by the European Union's Horizon 2020 research and innovation programme under Grant Agreement No. 820575 (TiPACCs). RW further acknowledges support by the European Union's Horizon
2020 research and innovation programme under Grant Agreement No. 869304 (PROTECT). Development of PISM is supported by NASA grants 20-CRYO2020-0052 and 80NSSC22K0274 and NSF grant OAC-2118285. The authors gratefully acknowledge the European Regional Development Fund (ERDF), the German Federal Ministry of Education and Research and the Land Brandenburg for supporting this project by providing resources on the high performance computer system at the Potsdam Institute for Climate Impact Research.

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

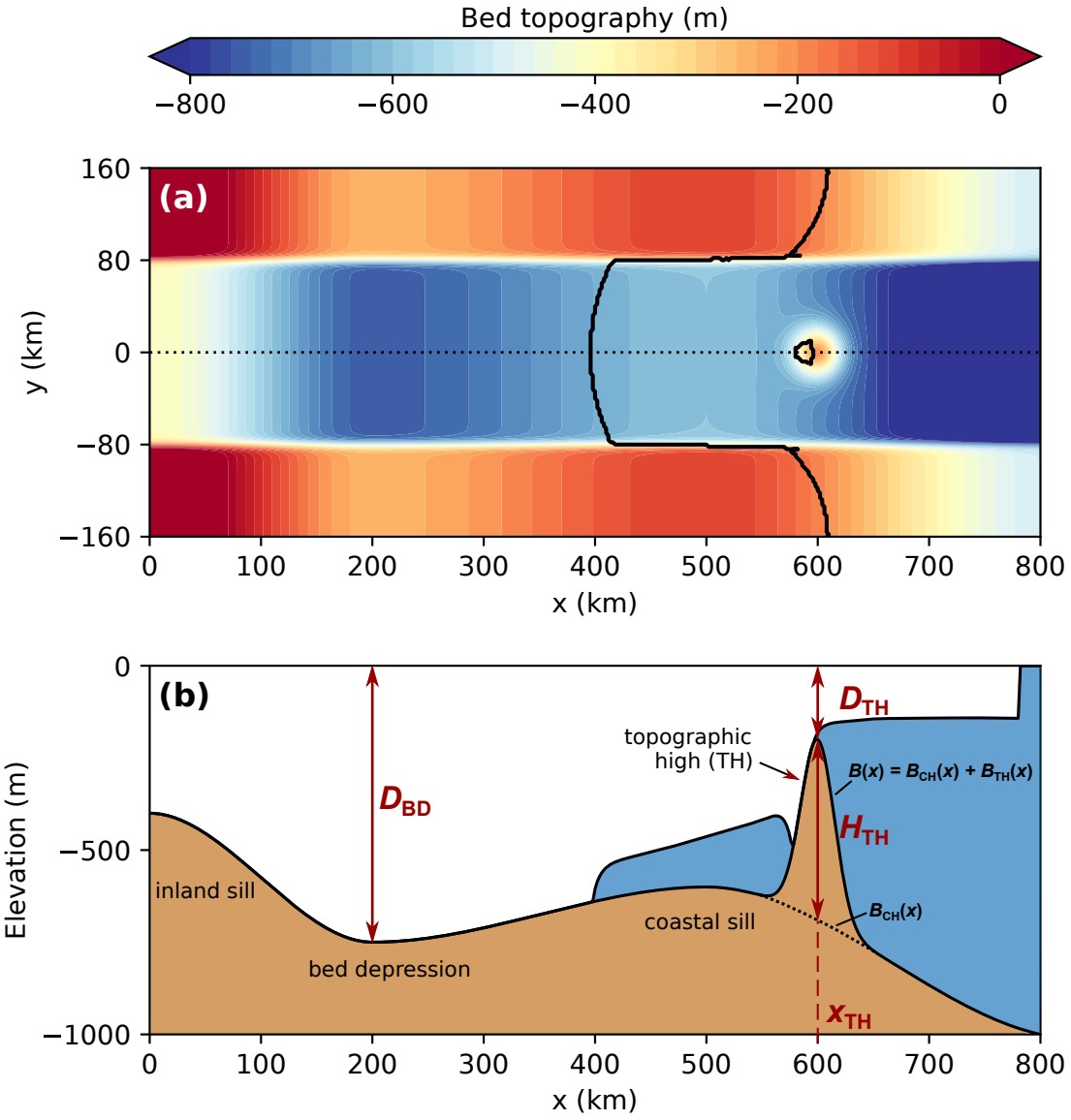

**Figure 1. (a)** Channel-type bed topography, $B(x,y)$, prescribed in the laterally confined simulations (colorbar) with steady-state GL position of the ice-sheet-shelf system and its pinning point (black contours). **(b)** Cross section along centerline ($y = 0$, dotted line in panel a) of region below sea level with bed topography (brown), ice body (white) and ocean (blue). The bed profile, $B(x)$ (continuous black contour), results from the superposition of the channel component of the bed topography, $B_{\mathrm{CH}}(x)$ (dotted black contour), and the Gaussian-shaped topographic high, $B_{\mathrm{TH}}(x)$, of height $H_{\mathrm{TH}}$ (see Eqs. 1-3). The parameters varied in the experiments are denoted in red, i.e., the depth of the bed depression, $D_{\mathrm{BD}}$, the depth of the topographic high, $D_{\mathrm{TH}}$, and the location of the topographic high, $x_{\mathrm{TH}}$. Also highlighted are the locations of the topographic high, the inland sill, the bed depression and the coastal sill as referred to in the text. For the example shown here, the parameter values are $D_{\mathrm{BD}} = 750$ m, $D_{\mathrm{TH}} = 200$ m and $x_{\mathrm{TH}} = 600$ km.

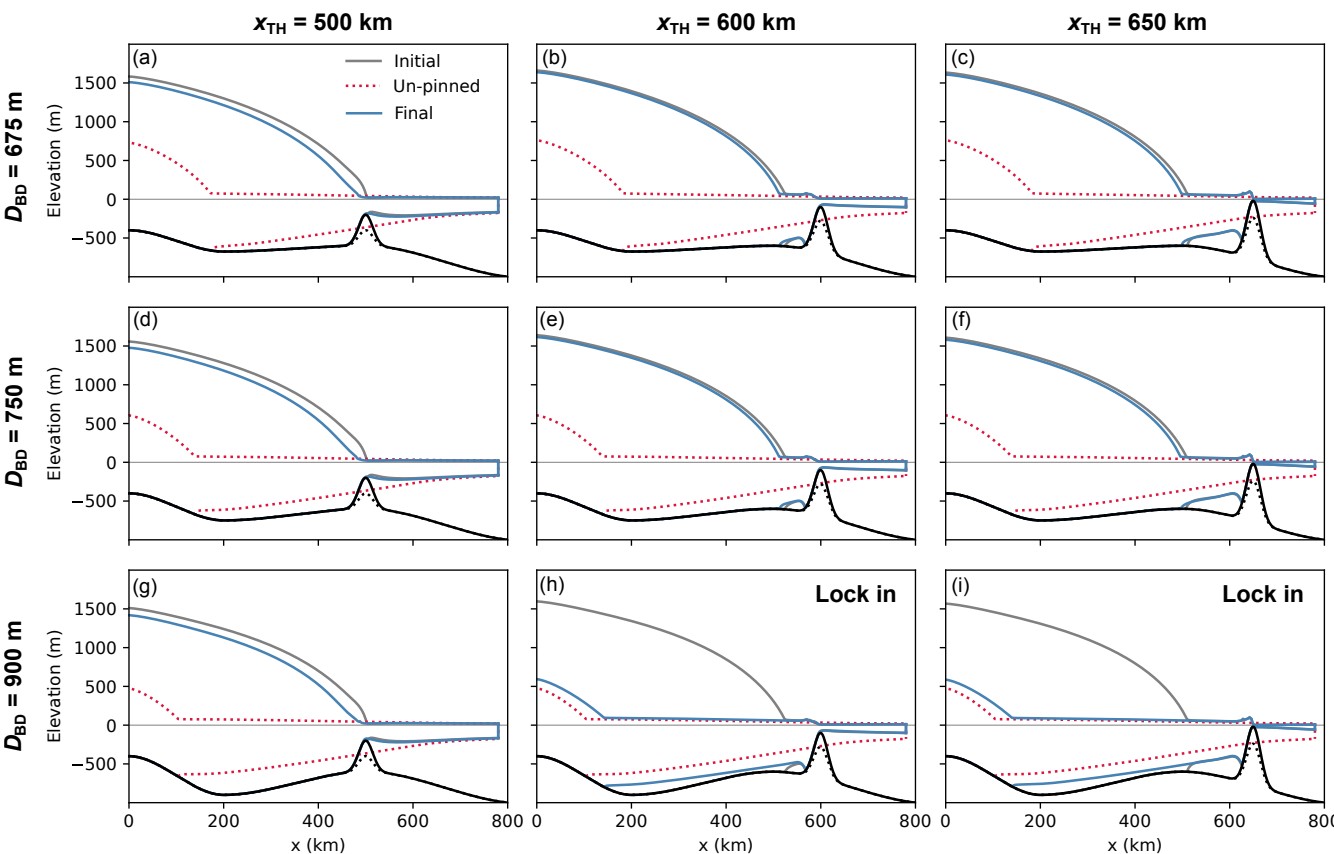

**Figure 2.** Steady-state centerline profiles for each set of the laterally confined hysteresis experiments, i.e., for each of the nine combinations of bed-depression depth, $D_{BD}$ (columns), and topographic-high location, $x_{TH}$ (rows). Shown are the initial state before perturbation (grey), the state after ice-shelf ungrounding from the topographic high (red dotted) and the final state, i.e., after the topographic high has been raised to its original value (blue). Bed topography in black with the dotted profile showing the topographic high at the stage of ice-shelf ungrounding (associated with red dotted glacier profile). The two sets of hysteresis experiments showing lock-in behavior (the glacier does not revert to its original state) are marked in the upper right corner.

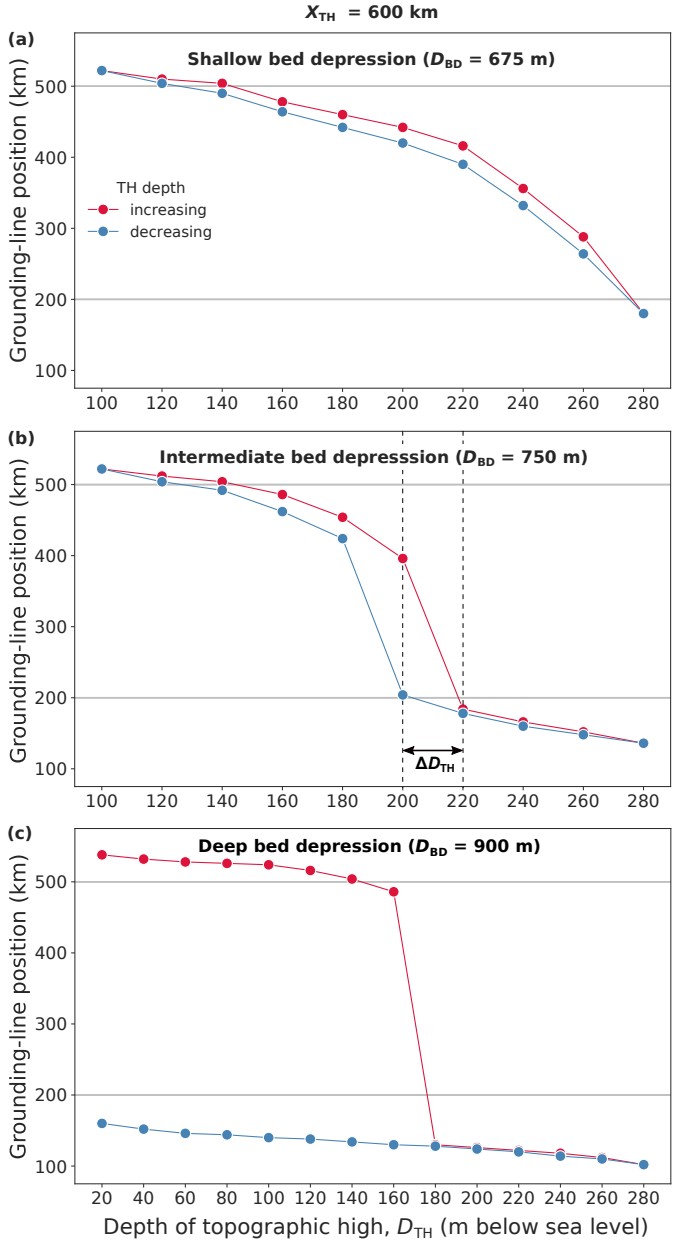

**Figure 3.** Hysteresis curves of steady-state centerline GL positions in the laterally confined simulations for a step-wise increase (red) and subsequent decrease (blue) of the depth of the topopgraphic high on which the ice shelf is pinned (see Fig. S1 for timeseries of the GL evolution). The topographic high is located at $x_{TH} = 600$ km. The panels show results for three different bed-depression depths, **(a)** $D_{BD} = 675$ m (shallow), **(b)** $D_{BD} = 750$ m (intermediate) and **(c)** $D_{BD} = 900$ m (deep). The two grey horizontal lines mark the range of the retrograde bed section between the tip of the coastal sill (at $x = 500$ km) and the deepest point of the bed depression (at $x = 200$ km). The dashed vertical lines in panel b denote the hysteresis gap $\Delta D_{TH}$, quantifying the amount of additional change in forcing required to tip back the glacier from its retreated state towards its advanced state on the path of re-growth (blue curve) compared to the path of retreat (red curve).

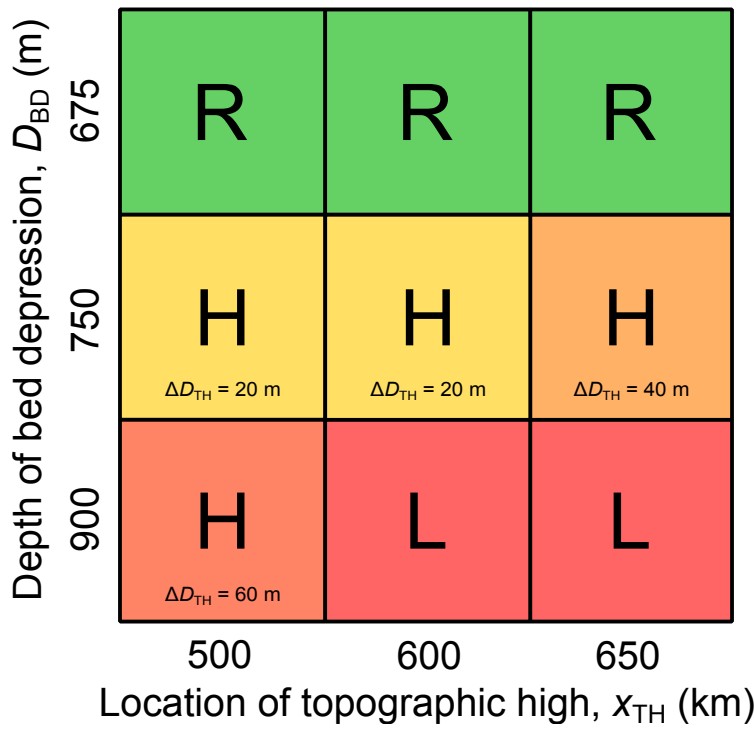

**Figure 4.** Regime diagram of qualitative glacier response in the laterally confined simulations to the applied lowering and subsequent raising of the topographic high on which the ice shelf is grounded, dependent on the depth of the bed depression below sea level ($x$ axis) and the distance of the topographic high to the ice divide ($y$ axis). Regimes are "R" for reversible glacier retreat and re-advance (green), "H" for hysteretic behavior (yellow to reddish; more reddish for larger hysteresis gap $\Delta h_{\mathrm{TH}}$ between retreat and advance) and "L" for lock in, i.e., no re-advance (red).

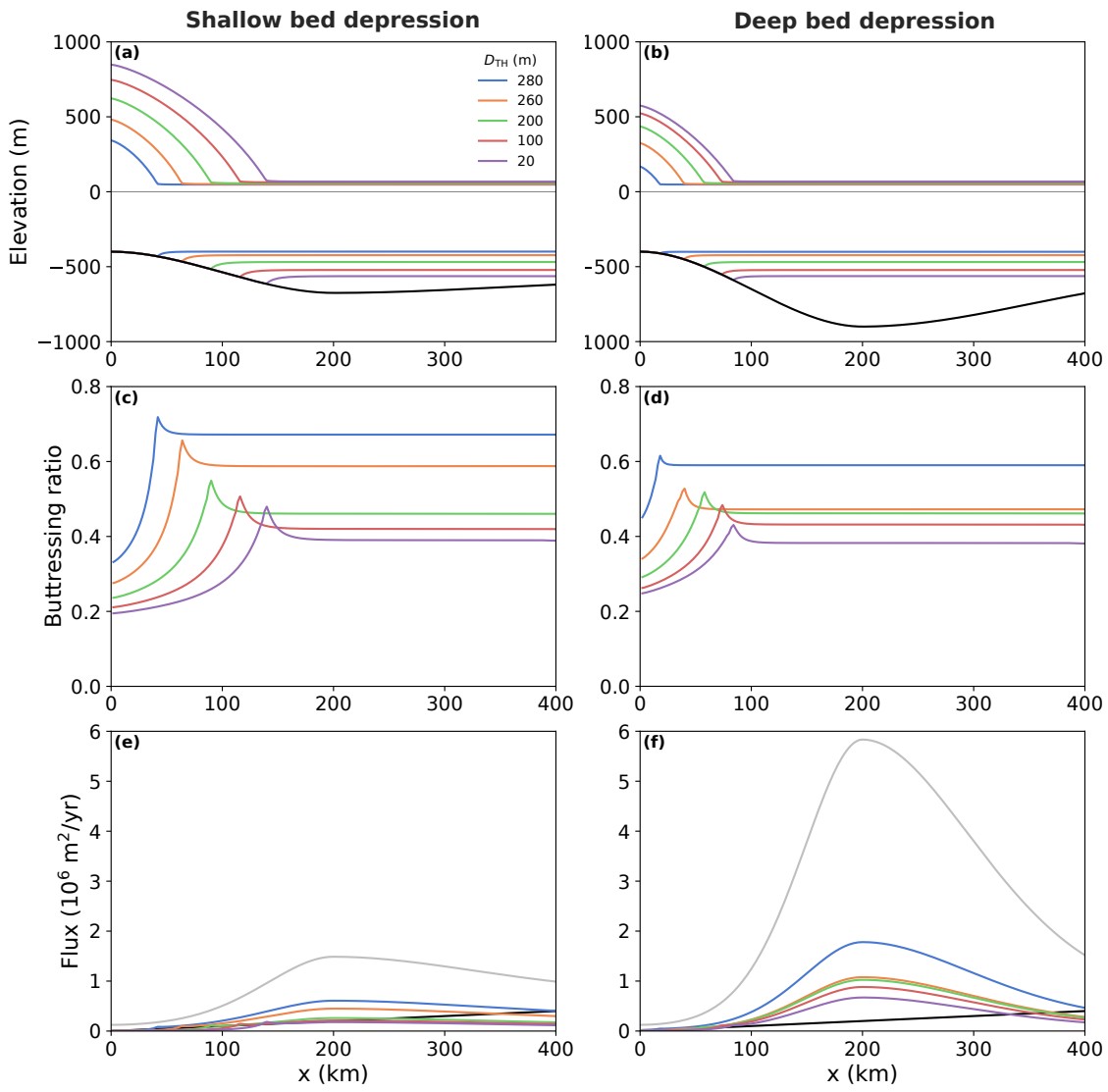

**Figure 5.** Centerline diagnostics from laterally unconfined simulations of the re-growing ice-sheet-shelf system (decreasing topographic-high depth $D_{TH}$; legend) for the cases of a shallow bed depression (left-hand-side column; $D_{BD} = 675$ m) versus a deep bed depression (right-hand-side column; $D_{BD} = 900$ m). **(a)**, **(b)** Profiles of the ice-sheet-shelf system (colors) and underlying bed topography (black). **(c)**, **(d)** Buttressing ratios diagnosed according to Gudmundsson et al. (2023, Eq. 6). **(e)**, **(f)** GL flux (colors) calculated according to the boundary layer theory by Schoof (2007, Eq. 29), based on the prescribed bed topography and the buttressing ratio. Integrated surface accumulation rate given by black line. The GL flux profile for the unbuttressed case is shown in grey.

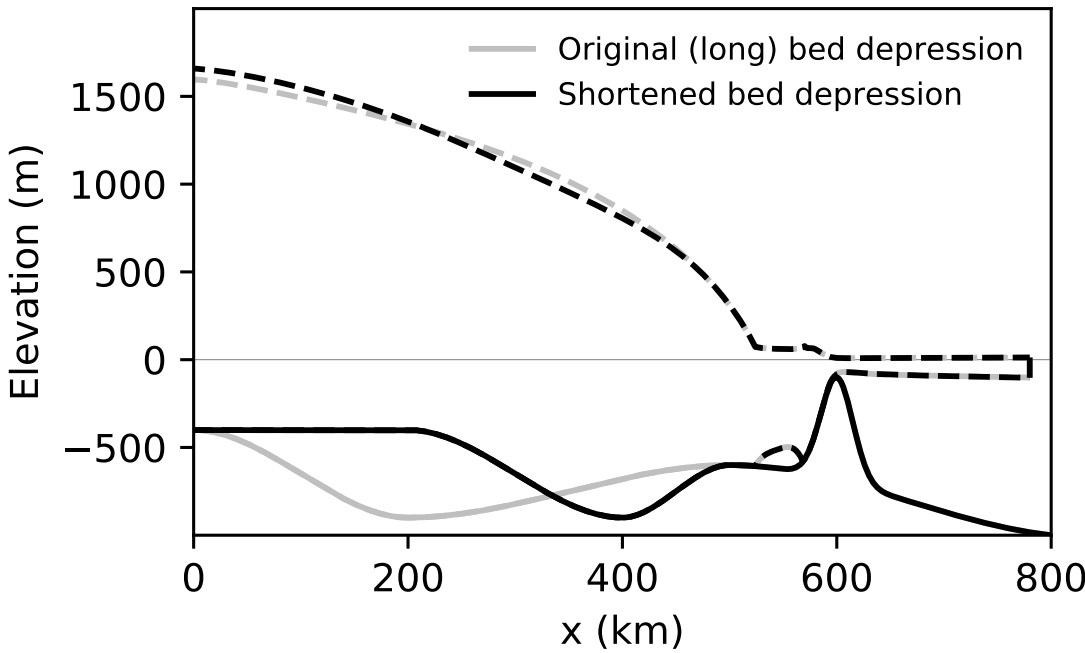

**Figure A1.** Bed geometry with a shortened bed depression (black, discussed in the Appendix) versus the original setup (grey; discussed in the main text) along the centerline ($y = 0$) of the setup. Bed profiles are continuous and associated steady-state glacier profiles prior to unstable retreat are dashed. For the example shown here, the parameter values are $D_{\mathrm{BD}} = 900$ m, $D_{\mathrm{TH}} = 100$ m and $x_{\mathrm{TH}} = 600$ km.

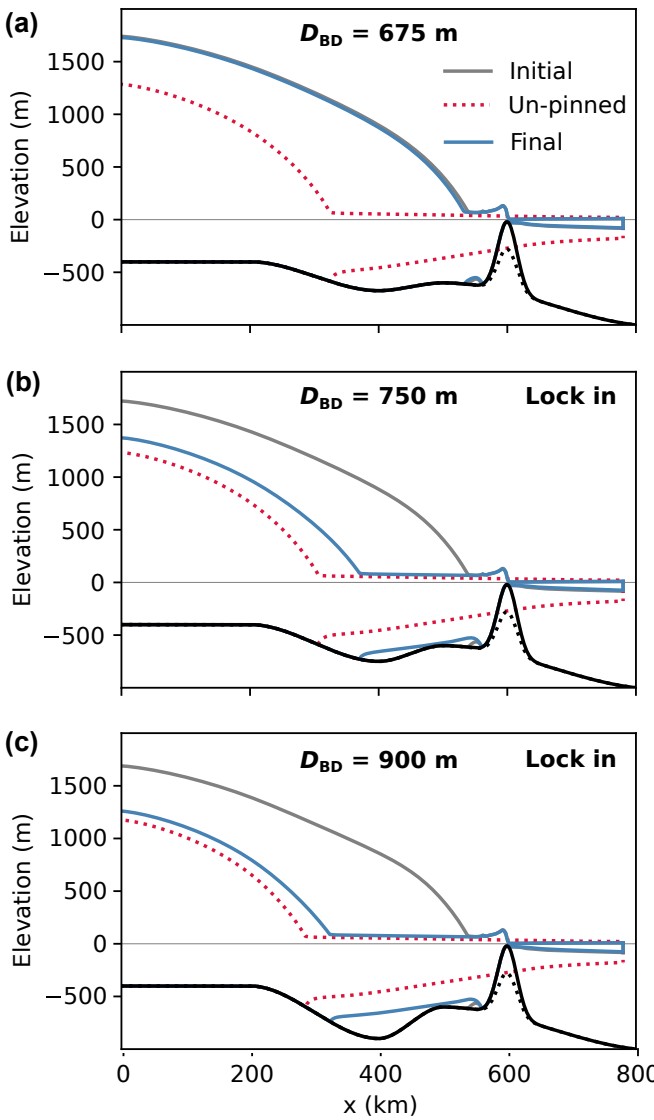

**Figure A2.** Steady-state centerline profiles analogous to Fig. 2 for the laterally confined experiments prescribing a shortened bed depression. (a) Shallow, (b) intermediate and (c) deep bed depression. The topographic high is located at $x_{\mathrm{TH}} = 600$ km.

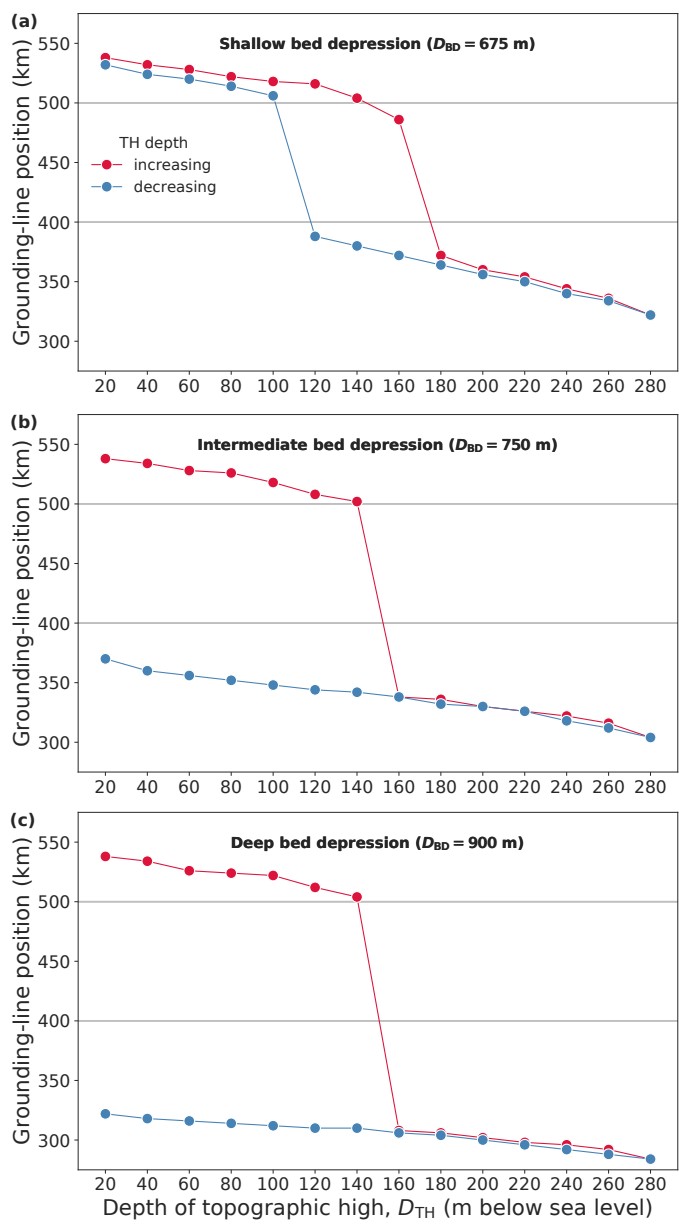

**Figure A3.** Hysteresis curves of the laterally confined simulations analogous to Fig. 3 for the shortened bed depression and a topographic-high location of $x_\mathrm{TH} = 600$ km. The two grey horizontal lines mark the range of the retrograde bed section between the tip of the coastal sill (at $x = 500$ km) and the deepest point of the bed depression (at $x = 400$ km).

**Table 1.** Physical constants and parameter values as prescribed in the laterally confined and laterally unconfined simulations.

| Parameter | Value | | Unit | Physical meaning |
|---|---|---|---|---|
| | Laterally confined | Laterally unconfined | | |
| $a$ | 0.15 | 1.0 | m yr$^{-1}$ | Surface accumulation rate |
| $A$ | $3.169 \cdot 10^{-25}$ | | Pa$^{-3}$ s$^{-1}$ | Ice softness, related to ice temperature $T$ |
| | | | | via Arrhenius law (Glen, 1955) |
| $T$ | $-13.2$ | | °C | Ice temperature |
| $C$ | $3.981 \cdot 10^{6}$ | | Pa m$^{-1/3}$ s$^{1/3}$ | Basal friction parameter, |
| | | | | entering Eq. (3) of Cornford et al. (2020) |
| $g$ | 9.81 | | m s$^{-2}$ | Gravitational acceleration |
| $m$ | 1/3 | | | Basal friction exponent, |
| | | | | entering Eq. (3) of Cornford et al. (2020) |
| $n$ | 3 | | | Exponent in Glen's law |
| $\rho_i$ | 918 | | kg m$^{-3}$ | Density of ice |
| $\rho_o$ | 1028 | | kg m$^{-3}$ | Density of ocean water |
| $d_c$ | 500 | - | m | Depth of bed trough compared with side walls, |
| | | | | entering Eq. (1) of Cornford et al. (2020) |
| $f_c$ | 4 | | km | Characteristic width of bed-trough side walls, |
| | | | | entering Eq. (1) of Cornford et al. (2020) |
| $w_c$ | 80 | 160 | km | Half-width of bed trough, |
| | | | | entering Eq. (1) of Cornford et al. (2020) |
| $D_{\mathrm{BD}}$ | $\{675, 750, 900\}$ | $\{675, 900\}$ | m | Depth of bed depression below sea level |
| $D_{\mathrm{TH}}$ | $\{20, ..., 400\}$ | $\{20, ..., 300\}$ | m | Range of topographic-high depths (20-m steps) |
| $L_x$ | 800 | | km | Length of computational domain ($x$ dimension) |
| $L_y$ | 320 | 160 | km | Width of computational domain ($y$ dimension) |
| $x_{\mathrm{CF}}$ | 780 | | km | Position of fixed calving front |
| $x_{\mathrm{TH}}$ | $\{500, 600, 650\}$ | 600 | km | $x$ location of topographic-high center |
| $y_{\mathrm{TH}}$ | 0 | | km | $y$ location of topographic-high center |