# Peer review of "Hysteresis of idealized, instability-prone outlet glaciers under variation of pinning-point buttressing"

_EGUsphere, 2024_

## Referee Comment (RC1)

**Review of Feldmann et al. "Hysteresis of idealized, instability-prone outlet glaciers under variation of pinning-point buttressing"**

April 9, 2024

**General comments:**

The manuscript by Feldman et al. presents a suite of idealised simulations that investigates the potential of hysteretic behaviour in response to variations in pinning-point buttressing. They find that the depth of the bathymetric depression as well as the height and distance of the pinning-point from the ice divide strongly influence the evolution of the outlet glacier and demonstrate that these variables can induce hysteretic behaviour. Based on the results from their idealised simulations, they then infer qualitative implications for real-world geometries in Antarctica.

I enjoyed reading this well-written, clearly structured, and well-illustrated paper. By investigating pinning-point buttressing, the authors address in my view a sometimes somewhat underappreciated topic that fits well within the scope of the Cryosphere (TC). I commend the authors for managing to produce a steady-state geometry that includes an ice rise. Overall, I think the paper is already in pretty good shape and I deem my comments minor. Therefore, I am in full support for publication in TC. I am listing below my comments that I would like the authors to take into consideration. I hope the authors find my comments helpful.

**Specific comments:**

1. I recommend to slightly restructure the "Methods" section. For once, I would move information about the grid resolution into 2.1. Then I would add the info whether the model is thermomechanically coupled or not (I believe not). If it is not, what kind of ice temperature is assumed? In section 2.2., I think I would appreciate a short mentioning of the dimensions of the computational domain. Then I would introduce a new section heading "2.3 Forcing and Boundary conditions" after line 115. This would basically

contain the paragraph starting in line 116. It would then be good to add what kind of lateral boundary conditions you apply e.g. no-slip, fixed calving front etc.

2. In your analysis of the Schoof flux formula (Eq. 4), you write this as a function of bed elevation at the grounding line B($x_{gl}$). In its original form, it is written as a function of ice thickness h($x_{gl}$). Do you use the flotation condition to get from one form to the other? And if you do, shouldn't there be a factor $\rho_o/\rho_i$ in front of B($x_{gl}$). I do not think, it affects your results, but this was unclear to me.

3. I consider this comment interesting but rather optional. You have looked at the effect of the depth of the bathymetric depression and the size and position of the pinning point. I wonder how much the length of the bathymetric depression matters? My suspicion is that you could have a deeper bathymetric depression if the length of the depression is shorter than in your current setup without inducing hysteretic behaviour. If it is not too difficult or time-consuming to run, I would be interested in such additional simulations. Especially considering that in the real world the bedrock topography is never as smooth as we make them in our models.

**Technical corrections:**

Title: I am not the biggest fan of the "instability-prone" phrase. My suggestion would be just to say "marine outlet glaciers"

Abstract:
L4: What is an Antarctic-type outlet glacier? I would call it a marine outlet glacier.

L5: Again instability-prone. How about "marine outlet glacier resting on a retrograde bed"?

L5: successive $->$ step-wise?

L8: delete "from"

L9: Whenever I read "collapsed", I think the glacier has disappeared. But other than in your unconfined simulations, I would rather call it "a retreated state" as the ice stream is still present, just not as advanced as before. This pretty much applies throughout the manuscript.

L25: Check correct spelling of MacAyeal citation

L29: Appreciate the citation, but it should really be the Schannwell et al. 2019 TC paper.

L40–50: Somewhere here, a reference to this new paper by Miles & Bingham 2024 in Nature might be worth adding.

L60: conceptual − > idealised?

L87 Eq. 2: How did you decide on the radius of you Gaussian bump? Any particular motivation?

L92: Since you only have three categories, maybe rename your "moderate" scenario to "intermediate"?

L99: Here and throughout, I would prefer if you used "ice sheet-ice shelf system" instead of "ice sheet-shelf system".

L106: "until changes in the glacier volume become negligible". Can you be more precise what your stopping criterion is?

L108: subsequently − > repeatedly

L108: "The perturbation is then reversed" − > "The sign of the perturbation is then reversed"

L114: I think somewhere here, I would mention explicitly that in your approach you decrease pinning-point buttressing through the reduction in contact area between ice shelf and topographic high. Because other strategies would also be possible.

L128: "step-wise elevation" − > "step-wise rise in elevation"

L153: "glacier tips" − > "glacier transitions"

L154–155: This is confusing. Is the ice shelf now grounded on the topographic high or not? Please clarify.

L166–174: When you cut out your domain, what are you boundary conditions at the lateral walls? Parallel ice velocity? Please add.

L210: Delete second "the"

L290: "it's" − > "its"

L302: "In real world" − > "In the real world"

Comment hyphenation: I noticed that you for example write "regrowth" but "re-advance". I am myself unsure what TC's policy here is, but it is probably a good idea to do this type of hyphenation consistently.

**Figures:**

The Figures are well illustrated and of very good quality. I only have a single tiny comment.

Fig. S4: Could you add the location of the topographic high to the plot as you did for Fig. S3 and Fig. S2.

Sincerely, Clemens Schannwell

---

## Referee Comment (RC2)

**Review of "Hysteresis of idealized, instability-prone outlet glaciers under variation of pinning-point buttressing", by Feldmann et al., 2024.**

**Summary**

This manuscript addresses the stability of ice streams or outlet glaciers in the presence of buttressing via topographic pinning points. The authors investigate this by conducting numerical experiments on a variety of idealized ice sheet configurations, thereby highlighting the hysteresis of such systems and the dependencies on topography. While hysteresis and the effects of buttressing on grounding line-stability have been examined in different configurations, this study adds an important consideration which is the buttressing from isolated pinning points. As they show, this has some effects that appear qualitatively different from buttressing via confinement. As such, I think this study can make a useful contribution to our general understanding of ice sheet stability, in particular by considering an element of geometric complexity (pinning points) that is difficult to incorporate into simpler theoretical or 1-D frameworks. They do this using a comprehensive 2D model (PISM) in an idealized geometry and straightforward experimental protocol, so that results can be illustrated within conceptual frameworks (i.e., hysteresis loops). Overall, this approach is sound, the model used is well-suited, and the figures are clear.

      However, I think there are some significant clarifications needed before publication. My major comments mainly deal with experimental choices that I think are incompletely explained, as well as the applicability to Thwaites glacier, which I feel is stretched (and somewhat inconsistent with the caveats the authors do mention). These are mainly issues of presentation and discussion, which I think are important to address, but they are not major issues with the experiments themselves.

      I also have a number of minor comments, which are mainly issues of clarity, and a few minor technical changes.

**Major comments to the authors**

**1) Connection to Thwaites and overall framing:** You are careful to mention caveats in the discussion section, and highlight the idealized nature of the setup, but I still find the connections drawn to Thwaites and Pine Island glaciers strained. This is not to say that no insights can be drawn – for example raising Thwaites and Pine Island as examples where lateral confinement and thus the likely role of pinning points is different. However, given the significant difference of the simulated bed slopes, overall bed depth, size of pinning points, etc... I am not sure why Thwaites is highlighted as the main real-world example. For example, I think it is a stretch to have Thwaites as a highlighted implication at the end of the abstract, given the actual experimental geometries.

      I think the insights in your study might be better appreciated if it were framed as a more general, theoretical contribution to understanding ice sheet stability and hysteresis. For one example, I think there are interesting implications for how ice sheets expand from a collapsed state and the role pinning points may play in that. You touch on this a bit when describing the "growth" branch of the hysteresis curve, but you could potentially expand on

it as a discussion point. Or, perhaps the geometries are more suitable for commenting on glacial/interglacial transitions across gentler continental shelf slopes? I am not saying you need to add these particular points, but I think diversifying the implications beyond current-day Thwaites might aid the overall robustness.

**Hysteresis analyzed via pinning point size vs. environmental forcings**. I think more explanation and justification should be provided for this choice. I'm not suggesting it isn't a valid choice to look at hysteresis across parameter space rather than an environmental forcing, and I realize you address the difference in the discussion. But given the potential effects (which you acknowledge, especially regarding basal melt), I think more explanation is needed. The reader is left wondering why this choice was made.

**Minor comments (line by line)**
- 48: would be helpful to provide some info from the Gudmundsson reference – it is a potentially significant qualifier in the context of this study.
- 49: "pinning points… vanishing" – ambiguous.. sounds like the topography is changing when I think you mean the pinning effect is vanishing as ice thins? Consider clarifying.
- 61: instead of "altering the buttressing strength", why not be more direct and say "by altering the amplitude of the pinning point"? There would be other ways to alter the buttressing strength of a given point (e.g., shelf thickness, rheology) so I think this would be clearer.
- 64: "simulated similar ice-sheet-shelf" is a long string of descriptors for "systems"… consider rephrasing to clarify
- 65: "local" presumably refers to the GL? Perhaps clarify.
- Fig. 2 and Generally: The upper limit for pinning point depth is extremely shallow. Might want to flag that for readers.
- 100: How is this elevation chosen a priori? Or do you mean this is chosen as the starting point for hysteresis experiments?
- 119: what is meant by "fast" ice dynamics, and why is it needed here? Fast in a rate sense, or as in "landfast ice"?
- 154-155: "… ice shelf remains pinned on the topographic high.  // Reversal of the perturbation leads to the re-grounding of the ice shelf …" … These two sentences seem contradictory. Does the ice shelf unground from the pinning point or not? Or does it depend on the experiment? This seems important as it implies some dependence on ice-shelf thickness after retreat, and therefore the boundary conditions and melt assumed (or lack of melt, in this case). In general, some more explanation of the mechanism of readvance might help the reader.
- 159: I would suggest a word other than "forcing", since the hysteresis experiments are done over a more abstract parameter space of bed topography. Maybe simply "perturbation"
- 188 (Flux balance analysis in general): A suggestion: it might be possible to consider these analyses for the confined case as well, using theoretical arguments for the buttressing factor as a function of shelf geometry. Haseloff and Sergienko (2018, J

Glac. doi:10.1017/jog.2018.30) derive such expressions that might allow you to expand this theoretical analysis to encompass more of your results.

- 225-227: does this imply the whole domain ungrounds into a uniform floating shelf pinned on the topographic high?
- 256: I'm not sure I understand this argument. It is stated in the beginning of the paragraph that readvance can't occur under these circumstances. Is the point that the flux analysis suggests that a shallower bed would facilitate flux balance in the confined case and therefore explains why readvance can occur in some of those simulations? That is reasonable but just follows from the dependence of GL flux on bed depth, so I'm not sure what is gained by trying to connect the flux analysis on the unconfined case back to the confined cases. I do in general think the flux-balance arguments are helpful for building physical intuition and connecting to theory.. but I think this paragraph needs to be clarified.
- 260: I think this summary sentence needs to be adjusted to better fit the analyses presented here. First, I don't think "rapid" should be used as it could be conflated with transient response which is not emphasized in these equilibrium hysteresis experiments. I'm assuming you mean a steep change with respect to the parameter space of D_th. Either way, I suggest clarifiying. Secondly, what is meant by "relatively small buttressing reduction" ? Relative to what?
- 278: I'm not sure if the Schoof 2012 reference is most applicable here, (a) because as you note, it is the unbutressed case that Schoof 2012 analyzes. (b) in Schoof 2012 I think the case that allows stability on a reverse slope is when there is strong ablation near the grounding line, such that retreat can increase the integrated accumulation flux. If you are referring to a different result, maybe clarify – but it seems Gudmundsson et al (2012) or others investigating stability on reverse slopes *with* buttressing would be more germane to your analyses anyway.
- 349: I would say "suggest" rather than "highlight".. highlight to me implies the analyses focused on Thwaites in particular. But again, see major comment on relationship to Thwaites.

**Technical comments**
- 38: comma after Shelf (".. ice shelf, to the two largest")
- 61: suggest "**By** altering..."
- 113: reduction/increase → reduction or increase
- 179 and on: suggest just writing out "first... second.. third.." etc. rather than 1), 2), 3).
- 227: it's → its
- 260: on **a** retrograde bed
- 290: it's → its
- 294: suggest "laterally weakly confined outlet glaciers" → "outlet glaciers with weak lateral confinement"
- 295: hysteretic/lock-in ... recommend choosing one or rewording to avoid "/"
- 301: minimal-invasive → minimally invasive
- 302: in **the** real world
- 342: in **the** case

---

## Author Comment (AC1)

**Response to Referee#1 (Clemens Schannwell)**

We would like to thank Clemens Schannwell for his willingness to review our manuscript, the helpful comments and the constructive suggestions to improve the manuscript. We are glad for the Referee's very positive assessment of our study and are happy to hear that he would support the publication in TC. We will prepare a revised version of the manuscript, addressing the points raised by the Referee. Please find below the *Referee's comments in italics* and our response in blue.

Sincerely,
Johannes Feldmann et al.

**Referee #1 (Clemens Schannwell)**

**General comments:**
*The manuscript by Feldmann et al. presents a suite of idealised simulations that investigates the potential of hysteretic behaviour in response to variations in pinning-point buttressing. They find that the depth of the bathymetric depression as well as the height and distance of the pinning-point from the ice divide strongly influence the evolution of the outlet glacier and demonstrate that these variables can induce hysteretic behaviour. Based on the results from their idealised simulations, they then infer qualitative implications for real-world geometries in Antarctica.*

*I enjoyed reading this well-written, clearly structured, and well-illustrated paper. By investigating pinning-point buttressing, the authors address in my view a sometimes somewhat underappreciated topic that fits well within the scope of the Cryosphere (TC). I commend the authors for managing to produce a steady-state geometry that includes an ice rise. Overall, I think the paper is already in pretty good shape and I deem my comments minor. Therefore, I am in full support for publication in TC. I am listing below my comments that I would like the authors to take into consideration. I hope the authors find my comments helpful.*

**Specific comments:**
*1. I recommend to slightly restructure the "Methods" section. For once, I would move information about the grid resolution into 2.1.*
Will be done.

*Then I would add the info whether the model is thermomechanically coupled or not (I believe not). If it is not, what kind of ice temperature is assumed?*
Will be done. The temperature value, which is related to the ice softness via an Arrhenius law, will be provided.

*In section 2.2., I think I would appreciate a short mentioning of the dimensions of the computational domain. Then I would introduce a new section heading "2.3 Forcing and Boundary conditions" after line 115. This would basically contain the paragraph starting in line 116. It would then be good to add what kind of lateral boundary conditions you apply e.g. no-slip, fixed calving front etc.*
Will be included.

*2. In your analysis of the Schoof flux formula (Eq. 4), you write this as a function of bed elevation at the grounding line B(xgl ). In its original form, it is written as a function of ice thickness h(xgl ). Do you use the flotation condition to get from one form to the other? And if you do, shouldn't there be a factor po/pi in front of B(xgl ). I do not think, it affects your results, but this was unclear to me.*

We are grateful for the Referee for discovering this inconsistency. Indeed, as assumed by the Referee, in our calculations we use the flotation criterion (factor -rho_o/rho_i) to translate between ice thickness and bed elevation at the grounding line. We simply forgot to include this constant when writing down the equations for the manuscript. We will correct this in the manuscript.

*3. I consider this comment interesting but rather optional. You have looked at the effect of the depth of the bathymetric depression and the size and position of the pinning point. I wonder how much the length of the bathymetric depression matters? My suspicion is that you could have a deeper bathymetric depression if the length of the depression is shorter than in your current setup without inducing hysteretic behaviour. If it is not too difficult or time-consuming to run, I would be interested in such additional simulations. Especially considering that in the real world the bedrock topography is never as smooth as we make them in our models.*

We very much acknowledge this idea and are currently running simulations to investigate the influence of the length of the bed depression, the results of which we might present in the paper Appendix.

***Technical corrections:***

*Title: I am not the biggest fan of the "instability-prone" phrase. My suggestion would be just to say "marine outlet glaciers"*

We understand the Referee's point here and are willing to follow the Referee's two following related suggestions (see next two comments). However, regarding the paper title we would be really in favor of using the term "instability-prone" as we are convinced that it describes the systems we model in an appropriate and concise way. It is important to us to state in the title that we simulate systems that represent not only marine outlet glaciers but those which are (theoretically) subject to the marine ice-sheet instabilty mechanism. Whilst using the the suggested term "marine outlet glaciers" alone would leave out an important part of information, adding more words like "resting on retrograde bed" would substantially lengthen the title and make it less readable. Since "instability-prone" involves all these details in one short term, we would like to refrain from removing this term from the title if the Editor is ok with it.

*Abstract:*
*L4: What is an Antarctic-type outlet glacier? I would call it a marine outlet glacier.*
*L5: Again instability-prone. How about "marine outlet glacier resting on a retrograde bed"?*
We will change the wording in the abstract according to the Referee's suggestion.

*L5: successive – > step-wise?*
Will be done.

*L8: delete "from"*
Will be done.

*L9: Whenever I read "collapsed", I think the glacier has disappeared. But other than in your unconfined simulations, I would rather call it "a retreated state" as the ice stream is still present, just not as advanced as before. This pretty much applies throughout the manuscript.*
Will be changed.

*L25: Check correct spelling of MacAyeal citation*
Will be done.

*L29: Appreciate the citation, but it should really be the Schannwell et al. 2019 TC paper.*
We apologize for mixing up the years here. Of course, it should be the paper from 2019 on ice-rise divide migration. Will be corrected.

*L40–50: Somewhere here, a reference to this new paper by Miles & Bingham 2024 in Nature might be worth adding.*
Thanks for the hint, we will include the reference!

*L60: conceptual – > idealised?*
Will be corrected.

*L87 Eq. 2: How did you decide on the radius of you Gaussian bump? Any particular motivation?*
The expression is adopted from Favier and Pattyn (2015). In fact, to keep things simple, we also adopted their value controlling the radius of the bump. We will mention this in the text.

*L92: Since you only have three categories, maybe rename your "moderate" scenario to "intermediate"?*
Will be done.

*L99: Here and throughout, I would prefer if you used "ice sheet-ice shelf system" instead of "ice sheet-shelf system".*
Thanks for the suggestion! In this case we really appreciate the conciseness of the term "ice-sheet-shelf system" and would prefer it over "ice sheet-ice shelf system", which is a bit more lengthy. In the end, this seems to be a matter of taste and we would suggest to leave it to the Editor to decide here.

*L106: "until changes in the glacier volume become negligible". Can you be more precise what your stopping criterion is?*
We will added more detail here.

*L108: subsequently – > repeatedly*
Will be corrected.

*L108: "The perturbation is then reversed" – > "The sign of the perturbation is then reversed"*
Will be done.

*L114: I think somewhere here, I would mention explicitly that in your approach you decrease pinning-point buttressing through the reduction in contact area between ice shelf and topographic high. Because other strategies would also be possible.*
We are grateful for this hint and will add a statement to the manuscript according to the Referee's suggestion.

*L128: "step-wise elevation" – > "step-wise rise in elevation"*
Will be changed.

*L153: "glacier tips" – > "glacier transitions"*
Will be changed.

*L154–155: This is confusing. Is the ice shelf now grounded on the topographic high or not? Please clarify.*
Thanks for pointing this out. We will add more detail to clarify.

*L166–174: When you cut out your domain, what are you boundary conditions at the lateral walls? Parallel ice velocity? Please add.*
Will be done.

*L210: Delete second "the"*
Will be done.

*L290: "it's" – > "its"*
Will be done.

*L302: "In real world" – > "In the real world"*
Will be done.

*Comment hyphenation: I noticed that you for example write "regrowth" but "re-advance". I am myself unsure what TC's policy here is, but it is probably a good idea to do this type of hyphenation consistently.*
We will change "regrowth" to "re-growth" throughout the manuscript for consistency.

**Figures:**

*The Figures are well illustrated and of very good quality. I only have a single tiny comment.*
*Fig. S4: Could you add the location of the topographic high to the plot as you did for Fig. S3 and Fig. S2.*
Will be done.

*Sincerely, Clemens Schannwell*

---

## Author Comment (AC2)

**Response to Referee#2 (John Erich Christian)**

We would like to thank John Erich Christian for the careful reading of our manuscript, for the helpful comments and suggestions as well as for the constructive criticism. We are delighted by the Referee's overall positive assessment of our study and will address all the points made by the Referee in a revised version of the manuscript. Please find below the *Referee's comments in italics* and our response in blue.

Sincerely,
Johannes Feldmann et al.

*Review of "Hysteresis of idealized, instability-prone outlet glaciers under variation of pinning-point buttressing", by Feldmann et al., 2024.*

*Summary*
*This manuscript addresses the stability of ice streams or outlet glaciers in the presence of buttressing via topographic pinning points. The authors investigate this by conducting numerical experiments on a variety of idealized ice sheet configurations, thereby highlighting the hysteresis of such systems and the dependencies on topography. While hysteresis and the effects of buttressing on grounding line-stability have been examined in different configurations, this study adds an important consideration which is the buttressing from isolated pinning points. As they show, this has some effects that appear qualitatively different from buttressing via confinement. As such, I think this study can make a useful contribution to our general understanding of ice sheet stability, in particular by considering an element of geometric complexity (pinning points) that is difficult to incorporate into simpler theoretical or 1-D frameworks. They do this using a comprehensive 2D model (PISM) in an idealized geometry and straightforward experimental protocol, so that results can be illustrated within conceptual frameworks (i.e., hysteresis loops). Overall, this approach is sound, the model used is well-suited, and the figures are clear.*

*However, I think there are some significant clarifications needed before publication. My major comments mainly deal with experimental choices that I think are incompletely explained, as well as the applicability to Thwaites glacier, which I feel is stretched (and somewhat inconsistent with the caveats the authors do mention). These are mainly issues of presentation and discussion, which I think are important to address, but they are not major issues with the experiments themselves. I also have a number of minor comments, which are mainly issues of clarity, and a few minor technical changes.*

*Major comments to the authors*
*1) Connection to Thwaites and overall framing: You are careful to mention caveats in the discussion section, and highlight the idealized nature of the setup, but I still find the connections drawn to Thwaites and Pine Island glaciers strained. This is not to say that no insights can be drawn – for example raising Thwaites and Pine Island as examples where lateral confinement and thus the likely role of pinning points is different. However, given the significant difference of the simulated bed slopes, overall bed depth, size of pinning points, etc... I am not sure why Thwaites is highlighted as the main real-world example. For example, I think it is a stretch to have Thwaites as a highlighted implication at the end of the abstract, given the actual experimental geometries.*
*I think the insights in your study might be better appreciated if it were framed as a more general, theoretical contribution to understanding ice sheet stability and hysteresis. For one example, I think there are interesting implications for how ice sheets expand from a collapsed state and the role pinning points may play in that. You touch on this a bit when describing the "growth" branch of the hysteresis curve, but you could potentially expand on it as a discussion point. Or, perhaps the geometries are more suitable for commenting on glacial/interglacial transitions across gentler continental shelf slopes? I am not saying you need to add these particular points, but I think diversifying the implications beyond current-day Thwaites might aid the overall robustness.*

We see the Referee's point here and are willing to give a broader picture of the implications of our results that is less focused on Thwaites Glacier. We appreciate the related suggestions made by the Referee, which we will try to pick up in the revised version of the manuscript, including a more detailed discussion of the effect of GIA (and thus pinning-point uplift) on ice-sheet advance and to which geometries our simulations apply in the real world.

*Hysteresis analyzed via pinning point size vs. environmental forcings. I think more explanation and justification should be provided for this choice. I'm not suggesting it isn't a valid choice to look at hysteresis across parameter space rather than an environmental forcing, and I realize you address the difference in the discussion. But given the potential effects (which you acknowledge, especially regarding basal melt), I think more explanation is needed. The reader is left wondering why this choice was made.*

We recognize that our purpose of applying a synthetic and simplified perturbation compared to a more realistic perturbation needs more explanation. As suggested by the Referee, we will provide a more detailed discussion of the synthetic perturbation versus environmeltal forcing for a better justification of our approach.

**Minor comments (line by line)**
*48: would be helpful to provide some info from the Gudmundsson reference – it is a potentially significant qualifier in the context of this study.*
We will provide more detail on this study as suggested by the Referee.

*49: "pinning points... vanishing" – ambiguous.. sounds like the topography is changing when I think you mean the pinning effect is vanishing as ice thins? Consider clarifying.*
We will improve the wording according to the Referee's suggestion.

*61: instead of "altering the buttressing strength", why not be more direct and say "by altering the amplitude of the pinning point"? There would be other ways to alter the buttressing strength of a given point (e.g., shelf thickness, rheology) so I think this would be clearer.*
We will modify this phrase for a clearer wording.

*64: "simulated similar ice-sheet-shelf" is a long string of descriptors for "systems"… consider rephrasing to clarify*
We will simplify the wording following the Referee's suggestion.

*65: "local" presumably refers to the GL? Perhaps clarify.*
We will clarify.

*Fig. 2 and Generally: The upper limit for pinning point depth is extremely shallow. Might want to flag that for readers.*
We will highlight this point.

*100: How is this elevation chosen a priori? Or do you mean this is chosen as the starting point for hysteresis experiments?*
The initial elevation of the topographic high is chosen as the starting point for the hysteresis experiment. We will add more a detailed explanation to clarify.

*119: what is meant by "fast" ice dynamics, and why is it needed here? Fast in a rate sense, or as in "landfast ice"?*
We will rephrased the sentence for clarification.

*154-155: "… ice shelf remains pinned on the topographic high. // Reversal of the perturbation leads to the re-grounding of the ice shelf …" … These two sentences seem contradictory. Does the ice shelf unground from the pinning point or not? Or does it depend on the experiment? This seems important as it implies some dependence on ice-shelf thickness after retreat, and therefore the boundary conditions and melt assumed (or lack of melt, in this case). In general, some more explanation of the mechanism of readvance might help the reader.*
We will rephrase this section to avoid irritation. We will go into more detail regarding the mechanism of GL readvance, as suggested by the Referee.

*159: I would suggest a word other than "forcing", since the hysteresis experiments are done over a more abstract parameter space of bed topography. Maybe simply "perturbation"*
Will be changed.

*188 (Flux balance analysis in general): A suggestion: it might be possible to consider these analyses for the confined case as well, using theoretical arguments for the buttressing factor as a function of shelf geometry.*

*Haseloff and Sergienko (2018, J Glac. doi:10.1017/jog.2018.30) derive such expressions that might allow you to expand this theoretical analysis to encompass more of your results.*

Thanks for the suggestion! In fact, when designing our study, we considered applying the theory by Haseloff and Sergienko, 2018 as it indeed involves an ice shelf and two horizontal dimensions. However, their approach is restricted to a single flow direction (x direction) and parameterizes buttressing through prescribing a lateral drag at the side margins of the ice shelf, not taking into account potential buttressing emerging from a pinning point. In constrast, our simulations involve both horizontal flow directions (x and y) and buttressing emerges inherently without prescribing any stresses. This limits the applicability of their theory to our simulations. Hence we took back a larger step, introducing the laterally unconfined simulations and apply Schoofs theory to these simulations. Please also see our response to the next but one comment.

*225-227: does this imply the whole domain ungrounds into a uniform floating shelf pinned on the topographic high?*
Yes, exactly. We will add this information explicitly to the text.

*256: I'm not sure I understand this argument. It is stated in the beginning of the paragraph that readvance can't occur under these circumstances. Is the point that the flux analysis suggests that a shallower bed would facilitate flux balance in the confined case and therefore explains why readvance can occur in some of those simulations? That is reasonable but just follows from the dependence of GL flux on bed depth, so I'm not sure what is gained by trying to connect the flux analysis on the unconfined case back to the confined cases. I do in general think the flux-balance arguments are helpful for building physical intuition and connecting to theory.. but I think this paragraph needs to be clarified.*

The point we want to make with the flux-balance analysis is to show the influence of both the bed depth AND the pinning point buttressing on the re-advance of the GL. Since we cannot apply the theory to the case of two-dimensional horizontal flow (our laterally unconfined simulations), we conducted the quasi-flowline simulations which still involve the buttressing effect of the pinning point. This allows us to explicitly demonstrate how the pinning-point buttressing increasingly suppresses the GL flux in response to the lifting of the topographic high. The presented outcome might not be surprising for an expert on this topic but we are convinced that our illustration will be quite valuable to the broader, less specialized readership. In any case, we think the flux-balance analysis is worth to be shown in the paper, as to our knowledge, it has not been presented this way before (involving pinning-point buttressing). We will revise the manuscript for a clearer discussion of the capabilities/limitations of our application of the flux-balance analysis.

*260: I think this summary sentence needs to be adjusted to better fit the analyses presented here. First, I don't think "rapid" should be used as it could be conflated with transient response which is not emphasized in these equilibrium hysteresis experiments. I'm assuming you mean a steep change with respect to the parameter space of D_th. Either way, I suggest clarifiying. Secondly, what is meant by "relatively small buttressing reduction" ? Relative to what?*

We thank the Referee for the careful reading. We will revise the sentence, as suggested by the Referee, replacing "rapid" by "large-scale" and "relatively small" by "step-wise".

*278: I'm not sure if the Schoof 2012 reference is most applicable here, (a) because as you note, it is the unbutressed case that Schoof 2012 analyzes. (b) in Schoof 2012 I think the case that allows stability on a reverse slope is when there is strong ablation near the grounding line, such that retreat can increase the integrated accumulation flux. If you are referring to a different result, maybe clarify – but it seems Gudmundsson et al (2012) or others investigating stability on reverse slopes with buttressing would be more germane to your analyses anyway.*

We thank the Referee for pointing this out. Indeed, the reference to Schoof is not appropriated here. We

will follow the Referee's suggestion and compare our results to studies using setups that are more similar to ours.

*349: I would say "suggest" rather than "highlight".. highlight to me implies the analyses focused on Thwaites in particular. But again, see major comment on relationship to Thwaites.*
Will be changed.

***Technical comments***
*- 38: comma after Shelf (".. ice shelf, to the two largest")*
*- 61: suggest "By altering…"*
*- 113: reduction/increase à reduction or increase*
*- 179 and on: suggest just writing out "first… second.. third.." etc. rather than 1), 2), 3).*
*- 227: it's à its*
*- 260: on a retrograde bed*
*- 290: it's à its*
*- 294: suggest "laterally weakly confined outlet glaciers" à "outlet glaciers with weak lateral confinement"*
*- 295: hysteretic/lock-in … recommend choosing one or rewording to avoid "/"*
*- 301: minimal-invasive à minimally invasive*
*- 302: in the real world*
*- 342: in the case*

We are glad for the Referee's technical corrections and will implement them in the revised version of the manuscript.

---

## Author Response (AR1)

**Response to the Editor**

Dear Dr. McCormack,

We would like to thank you for the careful handling of the review process. We are grateful for the valuable comments and suggestions from the referees as they really helped to improve our manuscript. Revising the manuscript, we took into account all the referees' suggestions (see our point-to-point answers below). This particularly involves clarifications and justifications requested by Dr. Christian regarding the experimental choices, the flux-balance analysis and the application of the results from our idealized simulations to the real world.

Furthermore, we carried out a set of additional simulations with a shorter length of the bed depression, as suggested by Dr. Schannwell. We present the results of these experiments in the Appendix (L433-474) to which we also added three new figures, i.e.,

- Fig. A1: a comparison of the modified to the original bed geometry along the centerline profile
- Fig. A2: steady-state profiles for the modified setup analogous to Fig. 2
- Fig. A3: hysteresis curves for the modified setup analogous to Fig. 3

We provide a brief discussion of the new results in Sect. 4 (L400-411).

Note that for consistency and better comparability we swapped the $x$ and $y$ axes in Figures 2 and 4 in order to align these figures with the hysteresis curves (Figs. 3, A3 and S1-S3) and also with Fig. A2. That is, in all the figures mentioned above, the value of the depth of the bed depression now consistently increases from top to bottom.

Once again, we would like to acknowledge the work and the time the Editor and the referees put into the review process from which we think that it really enriched the manuscript. We hope that the Editor and the referees are content with our proposed revisions to the manuscript. Please find below the *referees' comments in italics* and our detailed response in blue.

Sincerely,
Johannes Feldmann et al.

**Referee #1 (Clemens Schannwell)**

*Review of Feldmann et al. "Hysteresis of idealized, instability-prone outlet glaciers under variation of pinning-point buttressing"*

**General comments:**

*The manuscript by Feldmann et al. presents a suite of idealised simulations that investigates the potential of hysteretic behaviour in response to variations in pinning-point buttressing. They find that the depth of the bathymetric depression as well as the height and distance of the pinning-point from the ice divide strongly influence the evolution of the outlet glacier and demonstrate that these variables can induce hysteretic behaviour. Based on the results from their idealised simulations, they then infer qualitative implications for real-world geometries in Antarctica.*

*I enjoyed reading this well-written, clearly structured, and well-illustrated paper. By investigating pinning-point buttressing, the authors address in my view a sometimes somewhat underappreciated topic that fits well within the scope of the Cryosphere (TC). I commend the authors for managing to produce a steady-state geometry that includes an ice rise. Overall, I think the paper is already in pretty good shape and I deem my comments minor. Therefore, I am in full support for publication in TC. I am listing below my comments that I would like the authors to take into consideration. I hope the authors find my comments helpful.*

We would like to thank Dr. Schannwell for his willingness to review our manuscript, the valuable comments and the helpful suggestions to improve the manuscript. We are delighted by the Referee's very positive assessment of our study and are happy to hear that he would support the publication in TC. We gladly conducted the additional simulations proposed by the Referee, and think that the outcomes really enrich our study. We prepared a revised version of the manuscript, addressing the points raised by the Referee (see our point-to-point answers below).

**Specific comments:**

*1. I recommend to slightly restructure the "Methods" section. For once, I would move information about the grid resolution into 2.1.*

Done.

*Then I would add the info whether the model is thermomechanically coupled or not (I believe not). If it is not, what kind of ice temperature is assumed?*

It is indeed not thermomechanically coupled, which we now explicitly state in line 131. The temperature value is now given in Table 1 (third row). It is related to the ice softness via an Arrhenius law (now mentioned in second row of Table 1).

*In section 2.2., I think I would appreciate a short mentioning of the dimensions of the computational domain. Then I would introduce a new section heading "2.3 Forcing and Boundary conditions" after line 115. This would basically contain the paragraph starting in line 116. It would then be good to add what kind of lateral boundary conditions you apply e.g. no-slip, fixed calving front etc.*

We introduced this new section 2.3 (L130-142) where we mention all boundary conditions applied at the margins of the computational domain, following the Referee's suggestion. Note that we avoid prescribing a boundary condition at the ice divide by mirroring the setup at x=0, which we explain the text. The dimensions of the computational domain are now given in the same section.

*2. In your analysis of the Schoof flux formula (Eq. 4), you write this as a function of bed elevation at the grounding line B(xgl ). In its original form, it is written as a function of ice thickness h(xgl ). Do you use the flotation condition to get from one form to the other? And if you do, shouldn't there be a factor $\rho o/\rho i$ in front of B(xgl ). I do not think, it affects your results, but this was unclear to me.*

We are grateful for the Referee for discovering this inconsistency. Indeed, as assumed by the Referee, in our calculations we use the flotation criterion (factor -rho_o/rho_i) to translate between ice thickness and bed elevation at the grounding line. We simply forgot to include this constant when writing down the equations for the manuscript. It is now mentioned in L224 and part of the prefactor $c$ in Eq. (5).

*3. I consider this comment interesting but rather optional. You have looked at the effect of the depth of the bathymetric depression and the size and position of the pinning point. I wonder how much the length of the bathymetric depression matters? My suspicion is that you could have a deeper bathymetric depression if the length of the depression is shorter than in your current setup without inducing hysteretic behaviour. If it is not too difficult or time-consuming to run, I would be interested in such additional simulations. Especially considering that in the real world the bedrock topography is never as smooth as we make them in our models.*

We very much appreciate this idea to also investigate the influence of the length of the bed depression. Over the last weeks we thus carried out some more sets of simulations with a shortened bed depression. The results seem to be counter-intuitive at first sight, as a shorter bed depression (which is related to less absolute grounding-line retreat) leads to more pronounced irreversible/hysteretic behavior. The reason for this mainly lies in the fixed prescribed calving front position, which implies a lengthening of the ice shelf if the grounding line retreats: In case of a longer bed depression (and thus much stronger grounding-line retreat) the confined part of the ice shelf grows larger leading to more buttressing than in the short-depression case. This eventually facilitates glacier re-growth and thus reversibility when increasing the topogographic-high elevation. We present the results in the Appendix (L432-474, with three new figures A1-A3) and discuss them at the end of Sect. 4 (L400-411). We think that these additional simulations provide relevant information for ice-sheet modelers regarding the importance of the choice of the calving conditions or calving law applied in their simulations.

*Technical corrections:*

*Title: I am not the biggest fan of the "instability-prone" phrase. My suggestion would be just to say "marine outlet glaciers"*

We understand the Referee's point here and are willing to follow the Referee's two following related suggestions (see next two comments). However, regarding the paper title we would be really in favor of using the term "instability-prone" as we are convinced that it describes the systems we model in an appropriate and concise way. It is important to us to state in the title that we simulate systems that represent not only marine outlet glaciers but those which are (theoretically) subject to the marine ice-sheet instabilty mechanism. Whilst using the the suggested term "marine outlet glaciers" alone would leave out an important part of information, adding more words like "resting on retrograde bed" would substantially lengthen the title and make it less readable. Since "instability-prone" involves all these details in one short term, we would like to refrain from removing this term from the title if the Editor is ok with it.

*Abstract:*
*L4: What is an Antarctic-type outlet glacier? I would call it a marine outlet glacier.*
*L5: Again instability-prone. How about "marine outlet glacier resting on a retrograde bed"?*
We changed the wording in both instances according to the Referee's suggestion (L4-6).

*L5: successive – > step-wise?*
Done.

*L8: delete "from"*

Done.

*L9: Whenever I read "collapsed", I think the glacier has disappeared. But other than in your unconfined simulations, I would rather call it "a retreated state" as the ice stream is still present, just not as advanced as before. This pretty much applies throughout the manuscript.*
Changed throughout the manuscript, following the Referee's suggestion.

*L25: Check correct spelling of MacAyeal citation*
Done.

*L29: Appreciate the citation, but it should really be the Schannwell et al. 2019 TC paper.*
We apologize for mixing up the years here. Of course, it should be the paper from 2019 on ice-rise divide migration. Corrected.

*L40–50: Somewhere here, a reference to this new paper by Miles & Bingham 2024 in Nature might be worth adding.*
Thanks for the hint, we included the reference in L42-44 and L51-52.

*L60: conceptual – > idealised?*
Corrected.

*L87 Eq. 2: How did you decide on the radius of you Gaussian bump? Any particular motivation?*
The expression is adopted from Favier and Pattyn (2015). In fact, to keep things simple, we also adopted their value controlling the radius of the bump, which we mention in the text now (L91-92).

*L92: Since you only have three categories, maybe rename your "moderate" scenario to "intermediate"?*
Done.

*L99: Here and throughout, I would prefer if you used "ice sheet-ice shelf system" instead of "ice sheet-shelf system".*
Thanks for the suggestion! In this case we really appreciate the conciseness of the term "ice-sheet-shelf system" and would prefer it over "ice sheet-ice shelf system", which is a bit more lengthy. In the end, this seems to be a matter of taste and we would suggest to leave it to the Editor to decide here.

*L106: "until changes in the glacier volume become negligible". Can you be more precise what your stopping criterion is?*
As asked for by the Referee we added more detail here (L115).

*L108: subsequently – > repeatedly*
Corrected.

*L108: "The perturbation is then reversed" – > "The sign of the perturbation is then reversed"*
Done.

*L114: I think somewhere here, I would mention explicitly that in your approach you decrease pinning-point buttressing through the reduction in contact area between ice shelf and topographic high. Because other strategies would also be possible.*
We are grateful for this hint and added a sentence according to the Referee's suggestion (L123-125).

*L128: "step-wise elevation" – > "step-wise rise in elevation"*
Done.

*L153: "glacier tips" – > "glacier transitions"*
Done.

*L154–155: This is confusing. Is the ice shelf now grounded on the topographic high or not? Please clarify.*
Thanks for pointing this out. We added more detail to clarify (L173-180).

*L166–174: When you cut out your domain, what are you boundary conditions at the lateral walls? Parallel ice velocity? Please add.*
Done (L137-138).

*L210: Delete second "the"*
Done.

*L290: "it's" – > "its"*
Done.

*L302: "In real world" – > "In the real world"*
Done.

*Comment hyphenation: I noticed that you for example write "regrowth" but "re-advance". I am myself unsure what TC's policy here is, but it is probably a good idea to do this type of hyphenation consistently.*
We changed "regrowth" to "re-growth" throughout the manuscript for consistency.

**Figures:**

*The Figures are well illustrated and of very good quality. I only have a single tiny comment.*
*Fig. S4: Could you add the location of the topographic high to the plot as you did for Fig. S3 and Fig. S2.*
Done.

*Sincerely, Clemens Schannwell*

**Referee #2 (John Erich Christian)**

*Review of "Hysteresis of idealized, instability-prone outlet glaciers under variation of pinning-point buttressing", by Feldmann et al., 2024.*

*Summary*
*This manuscript addresses the stability of ice streams or outlet glaciers in the presence of buttressing via topographic pinning points. The authors investigate this by conducting numerical experiments on a variety of idealized ice sheet configurations, thereby highlighting the hysteresis of such systems and the dependencies on topography. While hysteresis and the effects of buttressing on grounding line-stability have been examined in different configurations, this study adds an important consideration which is the buttressing from isolated pinning points. As they show, this has some effects that appear qualitatively different from buttressing via confinement. As such, I think this study can make a useful contribution to our general understanding of ice sheet stability, in particular by considering an element of geometric complexity (pinning points) that is difficult to incorporate into simpler theoretical or 1-D frameworks. They do this using a comprehensive 2D model (PISM) in an idealized geometry and straightforward experimental protocol, so that results can be illustrated within conceptual frameworks (i.e., hysteresis loops). Overall, this approach is sound, the model used is well-suited, and the figures are clear.*

*However, I think there are some significant clarifications needed before publication. My major comments mainly deal with experimental choices that I think are incompletely explained, as well as the applicability to Thwaites glacier, which I feel is stretched (and somewhat inconsistent with the caveats the authors do mention). These are mainly issues of presentation and discussion, which I think are important to address, but they are not major issues with the experiments themselves. I also have a number of minor comments, which are mainly issues of clarity, and a few minor technical changes.*

We would like to thank Dr. Christian for the careful reading of our manuscript, for the helpful comments and suggestions as well as for the constructive criticism. We are glad for the Referee's overall positive assessment of our study. We address all the points made by the Referee in a revised version of the manuscript. In particular, the clarifications and more detailed explanations called for by the Referee greatly helped to improve our manuscript. Please find our point-to-point answers below.

*Major comments to the authors*
*1) Connection to Thwaites and overall framing: You are careful to mention caveats in the discussion section, and highlight the idealized nature of the setup, but I still find the connections drawn to Thwaites and Pine Island glaciers strained. This is not to say that no insights can be drawn – for example raising Thwaites and Pine Island as examples where lateral confinement and thus the likely role of pinning points is different. However, given the significant difference of the simulated bed slopes, overall bed depth, size of pinning points, etc... I am not sure why Thwaites is highlighted as the main real-world example. For example, I think it is a stretch to have Thwaites as a highlighted implication at the end of the abstract, given the actual experimental geometries.*
*I think the insights in your study might be better appreciated if it were framed as a more general, theoretical contribution to understanding ice sheet stability and hysteresis. For one example, I think there are interesting implications for how ice sheets expand from a collapsed state and the role pinning points may play in that. You touch on this a bit when describing the "growth" branch of the hysteresis curve, but you could potentially expand on it as a discussion point. Or, perhaps the geometries are more suitable for commenting on glacial/interglacial transitions across gentler continental shelf slopes? I am not saying you need to add these particular points, but I think diversifying the implications beyond current-day Thwaites might aid the overall robustness.*

We see the Referee's point here and revised the text to give a broader picture of the implications of our results that is less focused on Thwaites Glacier. Thus, as suggested by the Referee, we removed the last

sentence of the abstract, a specific conclusion on Thwaites Glacier which might not be justified by our set of experiments. Related to that, we also generalized our last statement in the conclusions (L428-429). Note that due to insightful findings from additional simulations suggested by the other Referee (presented in the Appendix and discussed in L400-412), we added to the end of the abstract a statement on the general role of the calving front for the (ir)reversiblity of grounding-line retreat (L14-15).

In addition to that, we now point out more clearly in the discussion section that we are not able to represent a specific glacier with our simulations (L333-335). Furthermore, we now give a more general classification of the retrograde bed slopes used in our experiments, explicitly mentioning their link to the more gentle bed slopes of the continental shelves beneath Ross and Filchner-Ronne ice shelves (L375-383). In this context, we also refer to potential of pinning points to facilitate GL re-advance, as suggested by the Referee (L383-389)

***Hysteresis analyzed via pinning point size vs. environmental forcings.*** *I think more explanation and justification should be provided for this choice. I'm not suggesting it isn't a valid choice to look at hysteresis across parameter space rather than an environmental forcing, and I realize you address the difference in the discussion. But given the potential effects (which you acknowledge, especially regarding basal melt), I think more explanation is needed. The reader is left wondering why this choice was made.*

We recognize that our purpose of applying a synthetic and simplified perturbation compared to a more realistic perturbation needs more explanation. As suggested by the Referee, we now provide a more detailed discussion of the synthetic perturbation versus environmeltal forcing and give a better reasoning for our approach (L342-354). We refer the reader to this discussion in the Methods section (L125-126).

***Minor comments (line by line)***
*48: would be helpful to provide some info from the Gudmundsson reference – it is a potentially significant qualifier in the context of this study.*
We now provide more detail on this study as suggested by the Referee (L48-52).

*49: "pinning points… vanishing" – ambiguous.. sounds like the topography is changing when I think you mean the pinning effect is vanishing as ice thins? Consider clarifying.*
We improved the wording according to the Referee's suggestion (L51-52).

*61: instead of "altering the buttressing strength", why not be more direct and say "by altering the amplitude of the pinning point"? There would be other ways to alter the buttressing strength of a given point (e.g., shelf thickness, rheology) so I think this would be clearer.*
We modified this phrase for a clearer wording (L63-65).

*64: "simulated similar ice-sheet-shelf" is a long string of descriptors for "systems"… consider rephrasing to clarify*
We simplified the wording following the Referee's suggestion.

*65: "local" presumably refers to the GL? Perhaps clarify.*
Done (L67).

*Fig. 2 and Generally: The upper limit for pinning point depth is extremely shallow. Might want to flag that for readers.*
We now give the range of the initial pinning-point depth in L112 of the Methods section and mention the shallowness of the initial topographic-high depths in the discussion (L306).

*100: How is this elevation chosen a priori? Or do you mean this is chosen as the starting point for hysteresis experiments?*
The initial elevation of the topographic high is chosen as the starting point for the hysteresis experiment. We added more detail to our explanation to clarify (L109-111).

*119: what is meant by "fast" ice dynamics, and why is it needed here? Fast in a rate sense, or as in "landfast ice"?*
We rephrased the sentence for clarification (now in L81-82).

*154-155: "... ice shelf remains pinned on the topographic high. // Reversal of the perturbation leads to the re-grounding of the ice shelf ..." ... These two sentences seem contradictory. Does the ice shelf unground from the pinning point or not? Or does it depend on the experiment? This seems important as it implies some dependence on ice-shelf thickness after retreat, and therefore the boundary conditions and melt assumed (or lack of melt, in this case). In general, some more explanation of the mechanism of readvance might help the reader.*
We rephrased this section to avoid irritation. We now go into more detail regarding the mechanism of GL readvance, as suggested by the Referee (L185-195).

*159: I would suggest a word other than "forcing", since the hysteresis experiments are done over a more abstract parameter space of bed topography. Maybe simply "perturbation"*
Done.

*188 (Flux balance analysis in general): A suggestion: it might be possible to consider these analyses for the confined case as well, using theoretical arguments for the buttressing factor as a function of shelf geometry. Haseloff and Sergienko (2018, J Glac. doi:10.1017/jog.2018.30) derive such expressions that might allow you to expand this theoretical analysis to encompass more of your results.*

Thanks for the suggestion! In fact, when designing our study, we considered applying the theory by Haseloff and Sergienko, 2018 as it involves the influence of the lateral extent of the ice shelf on buttressing. However, their approach is restricted to a single flow direction (x direction) and parameterizes buttressing through prescribing a lateral drag at the side margins of the ice shelf, not taking into account potential buttressing emerging from a pinning point. This limits the applicability of their theory to our simulations in which pinning-point buttressing emerges inherently from two-dimensional flow around the pinning point. Hence we took back a larger step, introducing the laterally unconfined simulations and applying the more general theory by Schoof 2007. We obtain the buttressing strength of the pinning point (entering Schoofs flux equation) from the stress field calculated in our model. Please also see our response to the next but one comment.

*225-227: does this imply the whole domain ungrounds into a uniform floating shelf pinned on the topographic high?*
Yes, exactly. We now added this information explicitly to the text (L255-256).

*256: I'm not sure I understand this argument. It is stated in the beginning of the paragraph that readvance can't occur under these circumstances. Is the point that the flux analysis suggests that a shallower bed would facilitate flux balance in the confined case and therefore explains why readvance can occur in some of those simulations? That is reasonable but just follows from the dependence of GL flux on bed depth, so I'm not sure what is gained by trying to connect the flux analysis on the unconfined case back to the confined cases. I do in general think the flux-balance arguments are helpful for building physical intuition and connecting to theory.. but I think this paragraph needs to be clarified.*

The point we want to make with the flux-balance analysis is to show the influence of both the bed depth AND the pinning point buttressing on the re-advance of the GL. Since we cannot apply the theory to the case of two-dimensional horizontal flow (our laterally unconfined simulations), we conducted the quasi-flowline simulations which still involve the buttressing effect of the pinning point. This allows us to explicitly demonstrate how the pinning-point buttressing increasingly suppresses the GL flux in response to the lifting of the topographic high. The presented outcome might not be surprising for an expert on this topic but we are convinced that our illustration will be quite valuable to the broader, less specialized readership. In any case, we think the flux-balance analysis is worth to be shown in the paper, as to our

knowledge, it has not been presented this way before (involving pinning-point buttressing).

We revised the paragraph mentioned by the Referee and now separated the actual results from the discussion of the capabilities/limitations of our application of the flux-balance analysis. Thus, at the end of the results section we now focus on the difference between the results of the laterally confined and unconfined simulations, respectively (L282-286). A reasoning for applying the flux-balance analysis according to Schoof 2007 is now given in the Discussion section (L319-327).

*260: I think this summary sentence needs to be adjusted to better fit the analyses presented here. First, I don't think "rapid" should be used as it could be conflated with transient response which is not emphasized in these equilibrium hysteresis experiments. I'm assuming you mean a steep change with respect to the parameter space of D_th. Either way, I suggest clarifiying. Secondly, what is meant by "relatively small buttressing reduction"? Relative to what?*

We thank the Referee for the careful reading. We revised the sentence, as suggested by the Referee, replacing "rapid" by "large-scale" and "relatively small" by "step-wise".

*278: I'm not sure if the Schoof 2012 reference is most applicable here, (a) because as you note, it is the unbutressed case that Schoof 2012 analyzes. (b) in Schoof 2012 I think the case that allows stability on a reverse slope is when there is strong ablation near the grounding line, such that retreat can increase the integrated accumulation flux. If you are referring to a different result, maybe clarify – but it seems Gudmundsson et al (2012) or others investigating stability on reverse slopes with buttressing would be more germane to your analyses anyway.*

We thank the Referee for pointing this out. Indeed, the reference to Schoof is not appropriated here and we thus removed it. The comparison to studies using a more similar setup (2 horizontal dimensions, buttressed ice sehlf) in terms of grounding-line stability on retrograde slopes is already done in L290-297.

*349: I would say "suggest" rather than "highlight".. highlight to me implies the analyses focused on Thwaites in particular. But again, see major comment on relationship to Thwaites.*
Done.

**Technical comments**
- *38: comma after Shelf (".. ice shelf, to the two largest")*
- *61: suggest "By altering..."*
- *113: reduction/increase à reduction or increase*
- *179 and on: suggest just writing out "first... second.. third.." etc. rather than 1), 2), 3).*
- *227: it's à its*
- *260: on a retrograde bed*
- *290: it's à its*
- *294: suggest "laterally weakly confined outlet glaciers" à "outlet glaciers with weak lateral confinement"*
- *295: hysteretic/lock-in ... recommend choosing one or rewording to avoid "/"*
- *301: minimal-invasive à minimally invasive*
- *302: in the real world*
- *342: in the case*

We are glad for the Referee's technical corrections and implemented all of them.

---

## Author Response (AR2)

**Response to the Editor**

Dear Dr. McCormack,

We would like to thank you for the quick reply and are delighted to read that the manuscript is likely not far from being accepted for publication after the minor revisions you mentioned. Please find below the *Editor's comments in italics* and our response in blue.

Sincerely,
Johannes Feldmann et al.

**Public justification (visible to the public if the article is accepted and published):**
*Dear Dr Feldmann and co-authors,*

*Thank you for your careful revision of the manuscript in light of the reviewers' comments.*

*Please find a couple of very minor specific comments below. After taking these into account, I'll have a look at the revised version again, but I strongly suspect the manuscript will be acceptable for publication in TC.*

*- I'm agree with you that "instability-prone" in the title and "ice sheet-shelf system" are both appropriate in the contexts you've applied them*
We are glad that we can leave the terms as they are.

*- There is now some repetition in the discussion around how including GIA would impact the results (lines 351-354 and 367-372). I suggest moving the earlier occurrence into the larger paragraph from line 367.*
*- L370: "However, also self-amplifying feedbacks..." --> "However, self-amplifying feedbacks also..."*
We shifted the first part discussing GIA to merge it with the second, as suggested by the Editor, and re-formulated a bit.

*Thanks again.*

*Best regards,*
*Felicity McCormack*